# VEGF/VEGFR2 signaling regulates hippocampal axon branching during development

Robert Luck[1,2,3†#], Severino Urban[1†#], Andromachi Karakatsani[1,2,3], Eva Harde[4,5,6], Sivakumar Sambandan[7‡], LaShae Nicholson[4,5,6], Silke Haverkamp[8§], Rebecca Mann[1], Ana Martin-Villalba[9], Erin Margaret Schuman[7], Amparo Acker-Palmer[4,5,6], Carmen Ruiz de Almodóvar[1,2,3]*

[1]Biochemistry Center (BZH), University of Heidelberg, Heidelberg, Germany; [2]European Center for Angioscience, Medicine Faculty Mannheim, Heidelberg University, Heidelberg, Germany; [3]Institute for Transfusion Medicine and Immunology, Medicine Faculty Mannheim, Heidelberg University, Heidelberg, Germany; [4]Institute of Cell Biology and Neuroscience, University of Frankfurt, Frankfurt am Main, Germany; [5]Neurovascular Interface group, Max Planck Institute for Brain Research, Frankfurt am Main, Germany; [6]Buchmann Institute for Molecular Life Sciences (BMLS), University of Frankfurt, Frankfurt am Main, Germany; [7]Department of Synaptic Plasticity, Max Planck Institute for Brain Research, Frankfurt am Main, Germany; [8]Imaging Facility, Max Planck Institute for Brain Research, Frankfurt am Main, Germany; [9]Department of Molecular Neurobiology, German Cancer Research Center (DKFZ), Heidelberg, Germany

*For correspondence:
carmen.ruizdealmodovar@
medma.uni-heidelberg.de

†These authors contributed equally to this work
#These authors are listed alphabetically

Present address: ‡Max Planck Institute for Biophysical Chemistry, Goettingen, Germany; §Department of Computational Neuroethology, Center of Advanced European Studies and Research (Caesar), Bonn, Germany

Competing interests: The authors declare that no competing interests exist.

**Abstract** Axon branching is crucial for proper formation of neuronal networks. Although originally identified as an angiogenic factor, VEGF also signals directly to neurons to regulate their development and function. Here we show that VEGF and its receptor VEGFR2 (also known as KDR or FLK1) are expressed in mouse hippocampal neurons during development, with VEGFR2 locally expressed in the CA3 region. Activation of VEGF/VEGFR2 signaling in isolated hippocampal neurons results in increased axon branching. Remarkably, inactivation of VEGFR2 also results in increased axon branching in vitro and in vivo. The increased CA3 axon branching is not productive as these axons are less mature and form less functional synapses with CA1 neurons. Mechanistically, while VEGF promotes the growth of formed branches without affecting filopodia formation, loss of VEGFR2 increases the number of filopodia and enhances the growth rate of new branches. Thus, a controlled VEGF/VEGFR2 signaling is required for proper CA3 hippocampal axon branching during mouse hippocampus development.

## Introduction

The correct function of neuronal networks relies on the establishment of specific patterns of connectivity between axons and dendrites. In this respect, proper regulation of axon branching during development is crucial to ensure precise axonal connections with multiple synaptic targets. Hippocampus development starts in mice around embryonic day E14 and continues through the first postnatal weeks (*Tole et al., 1997*). During this process, CA3 pyramidal neurons project their collateral axons to other CA3 neurons (associational connections) and to the CA1 hippocampal field via the Schaffer collaterals (*Witter, 2007*). During the first two postnatal weeks, CA3 axon collaterals undergo a process of growth, branching, remodeling and maturation to establish their connective

network (*Gomez-Di Cesare et al., 1997*). Although some molecular cues are described to regulate CA3 axon guidance (*Skutella and Nitsch, 2001*), the cues that regulate CA3 axon branching remain largely unknown.

De novo formation of axon branches, directly from the axon shaft (*Gallo, 2011*), comprises the major mechanism for establishing axon connectivity in the mammalian central nervous system (CNS) (*Kalil and Dent, 2014*). Target-derived cues such as extrinsic axon guidance cues (Netrin, Slits, Semaphorins or Ephrins), growth factors (Neurotrophins, BDNF, FGF-2, NGF) or morphogens (WNTs) can regulate axon branching by different dynamic strategies including promoting branch growth, repulsion, pruning or stabilization of branches (*Kalil and Dent, 2014*). Axon branching can also be regulated by activity-dependent mechanisms (*Yamamoto and López-Bendito, 2012*). In all cases, cytoskeleton reorganization is required for the initiation and growth of axon branches. Initiation of a new axon branch occurs when actin filaments accumulate along the axon (actin patches) and expand to membrane protrusions from which filopodia will emerge. Subsequently, microtubules will invade the filopodia thus giving rise to new axon branches (*Gallo, 2011*). Several of the molecular cues that regulate axon branching have been shown to regulate actin or microtubule dynamics. For example, Netrin-1 promotes axon branching in cortical neurons by inducing a rapid accumulation of actin filaments in filopodia (*Dent et al., 2004*). Also, asymmetric EGFR signaling has been shown to regulate actin dynamics and thereby axon branch pruning in *Drosophila* dorsal cluster neurons (*Zschätzsch et al., 2014*).

Vascular endothelial growth factor A (VEGFA, from here on termed VEGF) has been implicated in various neurodevelopmental processes including neurite outgrowth, neuronal survival and migration, as well as axon guidance (*Carmeliet and de Almodovar, 2013*; *Erskine et al., 2011*; *Meissirel et al., 2011*; *Ruiz de Almodovar et al., 2010*; *Ruiz de Almodovar et al., 2011*; *Schwarz et al., 2004*). Those direct effects on neurons are mediated by signaling via VEGFR2 (also known as KDR and FLK1) (*Carmeliet and de Almodovar, 2013*; *Erskine et al., 2011*; *Meissirel et al., 2011*; *Ruiz de Almodovar et al., 2010*; *Ruiz de Almodovar et al., 2011*; *Schwarz et al., 2004*) or via Neuropilin 1 (*Erskine et al., 2011*; *Schwarz et al., 2004*). Whether direct signaling of VEGF on neurons can regulate axon branching still remains unknown. Here we show that VEGF/VEGFR2 signaling regulates axon branching in CA3 hippocampal neurons. We find that VEGFR2 is expressed in CA3 hippocampal neurons during development and that VEGF is temporally and dynamically expressed in CA1-CA3 hippocampal neurons as well as in glial cells. We show that VEGFR2 is dynamically distributed along the axon and that VEGF stimulation increases VEGFR2 motility and localization towards actin-rich structures. We further show that CNS-specific VEGFR2 knockout mice display increased hippocampal axon branching in vivo, with branches that appear to be less mature and that form less functional synapses with CA1 neurons. Mechanistically, while VEGF stimulation results in increased axon branching by promoting the growth of newly formed branches in a Src Family Kinases (SFKs)-dependent manner, VEGFR2 inactivation leads to an increase in filopodia number that subsequently leads to increased branch formation.

## Results

### VEGFR2 and VEGF are expressed in the developing mouse hippocampus

Previous studies have demonstrated the expression of VEGF and its receptors in the adult murine hippocampus (*Licht et al., 2010*; *Wang et al., 2005*). To characterize their expression during hippocampal development, we performed in situ hybridization (ISH) at late embryonic (E18.5) and early postnatal (P4 and P8) stages. As expected, the mRNA encoding VEGFR2 was expressed in blood vessels (*Figure 1A*). In addition, we also detected VEGFR2 mRNA transcripts specifically in the CA3 hippocampal region throughout all developmental stages analyzed (*Figure 1A*). To further characterize the expression of VEGFR2, we took advantage of a transgenic knock-in mouse line in which GFP expression reliably reflects endogenous expression of VEGFR2 (*Vegfr2*-GFP, where exon 1 of the *Kdr* gene is replaced by GFP [*Ema et al., 2006*]). Immunostaining of postnatal brains at P4 and P8 with an antibody against GFP revealed specific labeling of the CA3 hippocampal region but not in the CA1, in addition to the strong labeling of endothelial cells (*Figure 1B–1D*, *Figure 1—figure supplement 1A*). These results indicate that expression of VEGFR2 mRNA is not only detected in

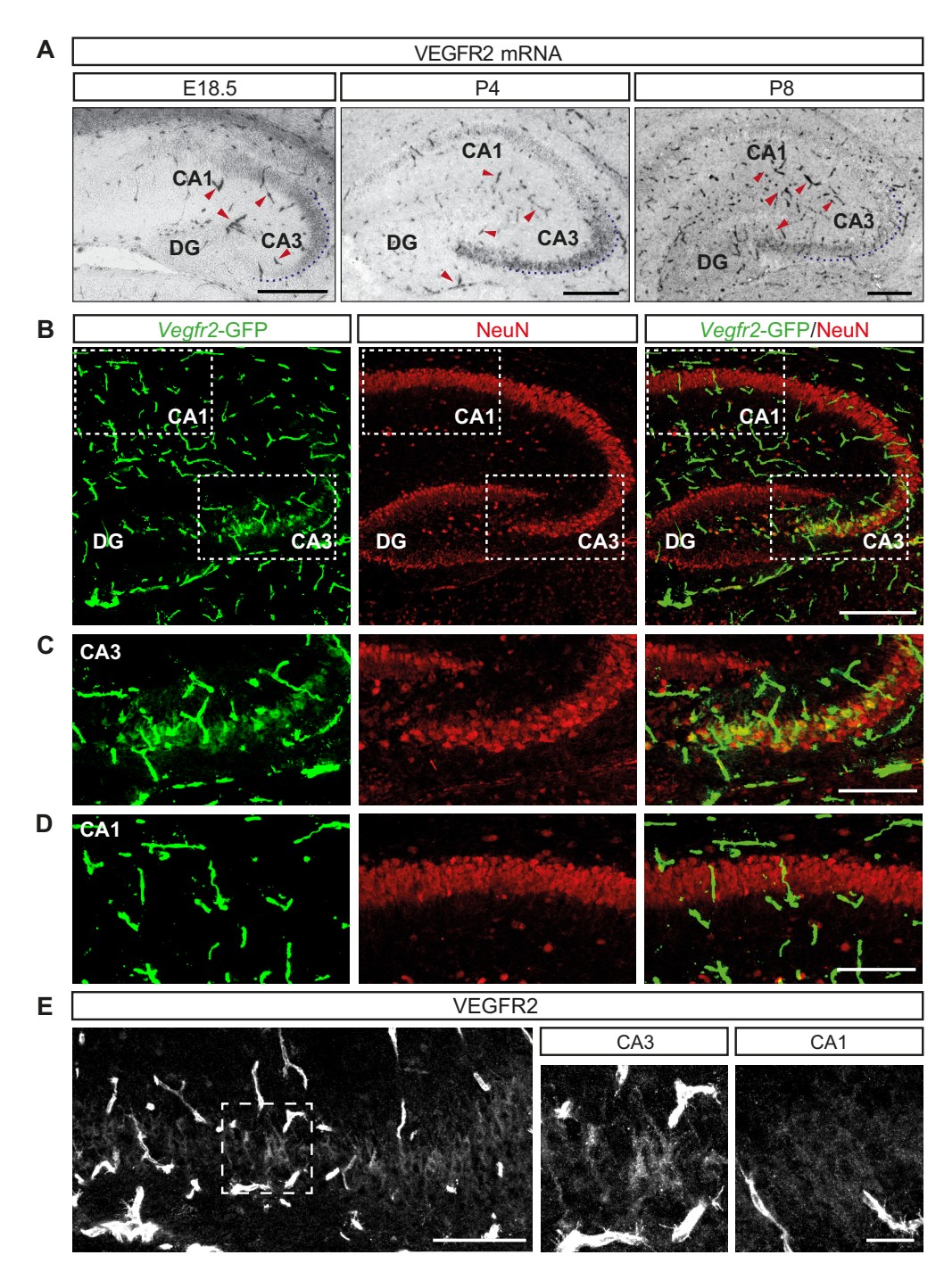

**Figure 1.** VEGFR2 is expressed by CA3 pyramidal neurons during hippocampal development. (**A**) VEGFR2 mRNA ISH in the hippocampus at E18.5, P4 and P8. Blood vessels are indicated by red arrowheads. Scale bars 250 µm. (**B**) GFP and NeuN immunostaining of *Vegfr2*-GFP hippocampus at P8. Scale bar 250 µm. (**C,D**) High magnification images of the CA3 region (**C**) and the CA1 region (**D**) from insets in (**B**). Scale bar 50 µm. (**E**) VEGFR2 immunostaining in the CA3 region of P8 hippocampus. High magnification images of the CA3 region (inset in left panel) and CA1 region are shown on the right. Scale bars 100 µm and 25 µm, respectively.

The online version of this article includes the following figure supplement(s) for figure 1:

**Figure supplement 1.** VEGFR2 expression in hippocampal neurons in vivo.

endothelial cells but also in cells of the CA3 region. In order to determine the CA3 cell types that express VEGFR2 mRNA we co-immunostained brain sections from P8 of *Vegfr2*-GFP mice with the pan-neuronal marker NeuN and the interneuron-specific marker calretinin. Analysis revealed that VEGFR2 is specifically expressed in pyramidal neurons, but not in interneurons (*Figure 1B–1D*; *Figure 1—figure supplement 1B*). This spatiotemporal analysis showed that VEGFR2 mRNA is expressed in CA3 pyramidal neurons during the first two postnatal weeks. As a third approach to characterize VEGFR2 protein localization, we performed immunostainings using a specific anti-VEGFR2 antibody previously used to detect VEGFR2 in both neurons and blood vessels (*Bellon et al., 2010*). Consistently, we detected VEGFR2 in CA3 pyramidal neurons at P8 (*Figure 1E*). Finally, RT-PCR and Western blotting of lysates from isolated primary hippocampal neurons verified VEGFR2 expression in vitro (*Figure 1—figure supplement 1D and E*). This developmental time window of VEGFR2 expression coincides with the dynamic neurite growth and branching of CA3 neurons.

VEGF (encoded by *Vegfa* in mouse) is the main ligand and activator of VEGFR2. To characterize the expression of VEGF we performed ISH for VEGF mRNA in the developing mouse hippocampus. We detected VEGF mRNA in the CA1, CA2 and CA3 hippocampal regions of all the embryonic and postnatal stages analyzed (*Figure 2A*). Additionally, VEGF mRNA was detected in the dentate gyrus (DG) during postnatal development (*Figure 2A*). Consistently, we detected VEGF protein in conditioned medium from cultured isolated primary hippocampal neurons (*Figure 2—figure supplement 1A*). At P8 VEGF mRNA expression was also detected in other cell types of the hippocampus, in addition to CA hippocampal neurons (*Figure 2A*). Recent publications identified endothelial cells as a cellular source of VEGF during CNS development (*Barber et al., 2018*; *Li et al., 2013*). To further identify the other cells types that express VEGF in the hippocampus, we combined ISH for VEGF mRNA with immunostainings for NeuN and the glial fibrillary acidic protein (GFAP), that specifically labels astrocytes. We observed that both pyramidal neurons and astrocytes of the developing CA hippocampal regions express VEGF mRNA (*Figure 2B and C*). Notably, the expression levels of VEGF mRNA decrease in the neuronal populations during postnatal development, but remain rather stable in the other cell types (*Figure 2A–2C*).

Taken together, our analysis indicates that VEGFR2 is expressed in pyramidal neurons of the CA3 area of the developing hippocampus. Additionally, VEGF is expressed in different cell populations of the developing hippocampus. This expression pattern opened the question of whether direct VEGF signaling to VEGFR2 in hippocampal neurons could regulate their development.

## VEGF induces axon branching in hippocampal neurons in vitro

As axon branching of CA3 hippocampal neurons coincides with the spatiotemporal expression of VEGFR2 within the first two postnatal weeks (*Gomez-Di Cesare et al., 1997*), we investigated whether VEGF/VEGFR2 signaling is important for axon branching. For this, we used cultures of primary hippocampal neurons and stimulated them with 100 ng/ml VEGF at day in vitro 1 (1 DIV). We subsequently analyzed the axonal morphology at 3 DIV (axons were defined as the longest neurite process [*Dotti et al., 1988*]). VEGF stimulation of hippocampal neurons resulted in a significant increase in both the number and length of axon branches, when compared with vehicle stimulated (control) neurons (*Figure 3A–3C*). The number and length of primary neurites (excluding the axon), as well as the axon length, were not affected upon VEGF stimulation (*Figure 3—figure supplement 1A–1C*). Next, we performed time-lapse video microscopy of 1 DIV hippocampal neurons to analyze the dynamics of axon branch formation. Analysis of the movies revealed that VEGF stimulation led to an increase in branching and net branch growth rate of axon branches, without affecting branch retraction (*Figure 3D and E*; *Figure 3—figure supplement 1D and E*; *Figure 3—video 1* and *Figure 3—video 2*).

Axon branches develop from filopodia that stabilize and continue elongating (*Kalil and Dent, 2014*). The first step in filopodia formation is the focal accumulation of actin filaments, known as actin patches (*Kalil and Dent, 2014*). Actin patches form spontaneously, grow in size and eventually dissipate. Only a subset of actin patches stabilizes, giving rise to protrusions and subsequently filopodia before fully dissipating (*Loudon et al., 2006*). To study these events in vitro, we transfected primary hippocampal neurons with a plasmid containing the calponin homology domain of the F-actin binding protein Utrophin (Utr-CH) fused to mCherry (mCherry-UtrCH) in order to visualize F-actin in the axon (*Burkel et al., 2007*) and performed time-lapse movies over the course of 2 to 10

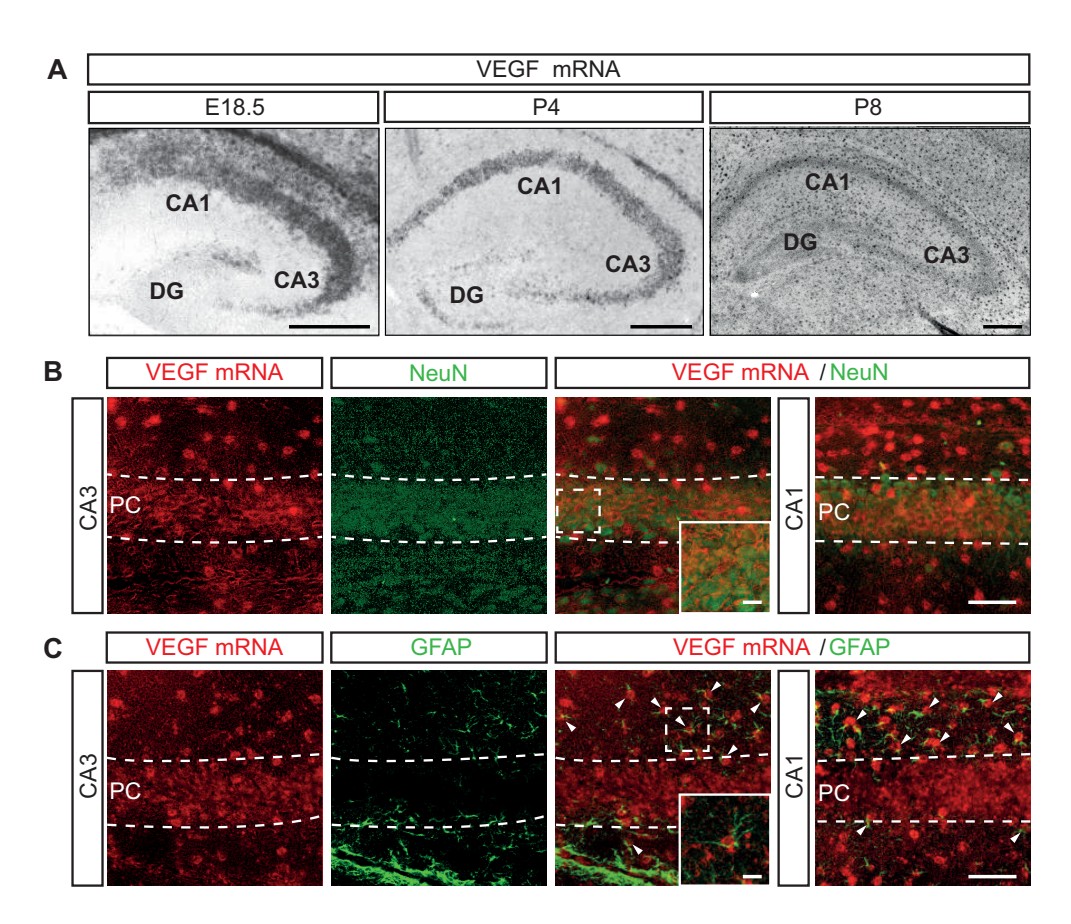

**Figure 2.** VEGF mRNA is expressed in different cell types during hippocampal development. (**A**) ISH for VEGF mRNA in the hippocampus at E18.5, P4 and P8. Scale bars 250 µm. (**B**) ISH for VEGF mRNA combined with NeuN immunostaining at P8. High magnification images of the CA1 and CA3 region are shown. VEGF-positive pyramidal cell layer (PC) is delineated by dotted lines. Scale bar 50 µm. (**C**) VEGF mRNA ISH combined with GFAP immunostaining at P8. Insets show high magnification images of the CA3 region (Scale bar 10 µm). The pyramidal cell layer (PC) is delineated by a dotted line. Arrowheads indicate VEGF mRNA positive glial cells. Scale bar 50 µm.

The online version of this article includes the following figure supplement(s) for figure 2:

**Figure supplement 1.** VEGF expression in hippocampal neurons in vitro.

min. Analysis of axonal segments showed that F-actin nucleation is mainly localized in patches along the axon shaft and branches. These patches appear and dissipate in a dynamic manner (*Figure 3— video 3*). Analysis of mCherry-UtrCH dynamics also revealed that membrane protrusions (<2 µm in length) arise from actin patches and can either transition into filopodia (>2 µm in length) or regress and lead to F-actin dissipation (*Figure 3F*; *Figure 3—video 4*). Using this experimental set up and quantifying the different events we show that stimulation of hippocampal neurons with VEGF did not result in any significant alteration in the total number of newly formed actin nucleation events (*Figure 3G*). Similarly, within the 10 min interval, the percentage of the newly formed protrusions and filopodia, as well as their size, was not affected upon VEGF stimulation (*Figure 3H and I*).

Altogether, these results indicate that VEGF stimulation of hippocampal neurons in vitro promotes the growth of newly formed branches, but does not affect the initial step of filopodia formation.

## VEGF induces VEGFR2 activation and motility and promotes axon branching in a Src family kinase dependent manner

To determine whether VEGF signals via VEGFR2 to control axon branching, we first determined the activation of VEGFR2 upon VEGF stimulation in our culture system. Indeed, VEGF stimulation of 1

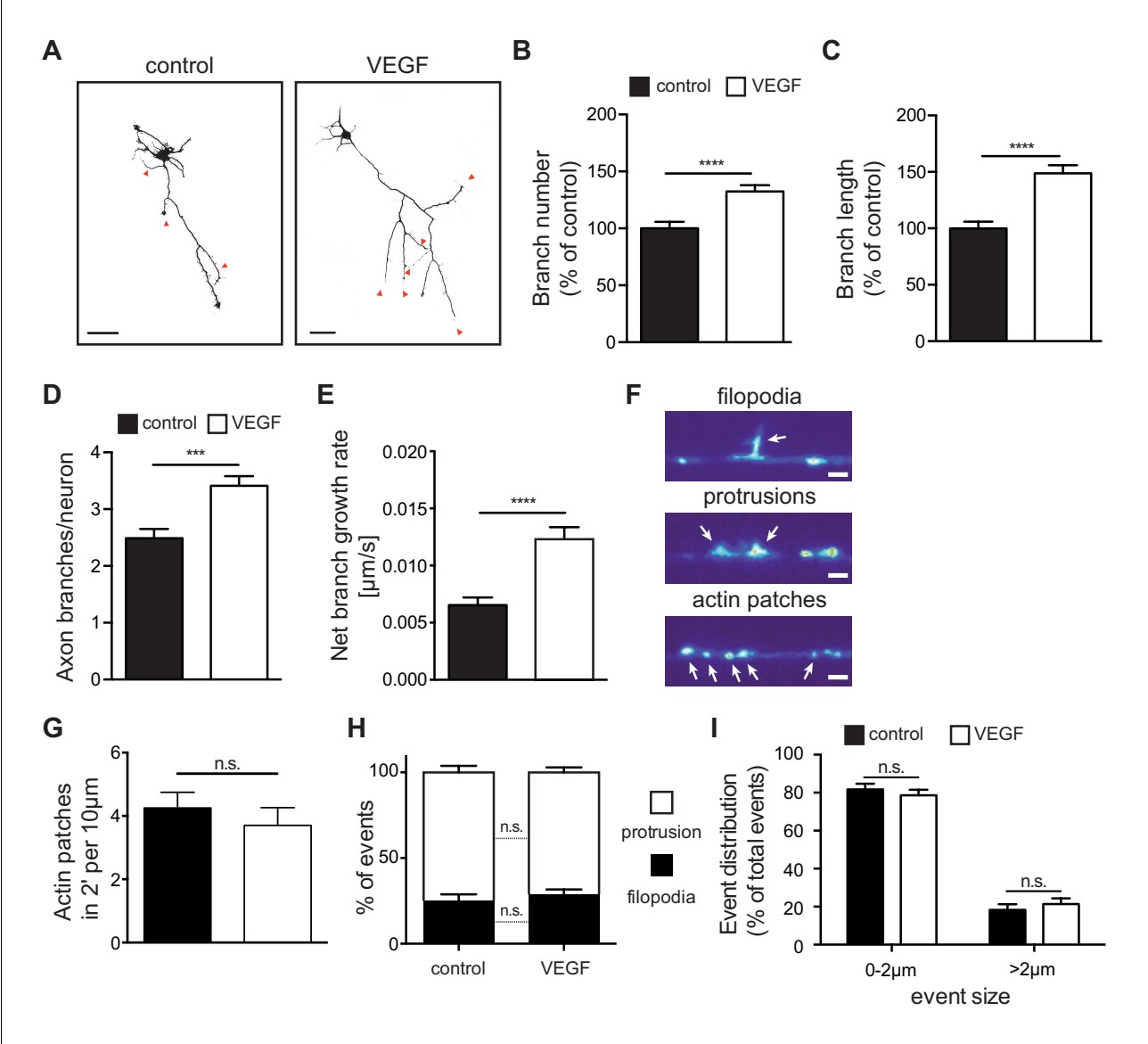

**Figure 3.** VEGF stimulation promotes the growth of axon branches in hippocampal neurons. (**A**) Representative images of 3 DIV hippocampal neurons stimulated with or without 100 ng/ml VEGF for 48 hr and stained with beta-III-tubulin. Scale bars 50 µm. (**B,C**) Quantification of axonal branch number (**B**) and branch length (**C**). Data are represented as % of non-stimulated control. Mean ± SEM,>60 neurons from n = 4. ****p<0.0001; unpaired Student's ttest. (**D,E**) 1 DIV hippocampal neurons were stimulated with 50 ng/ml VEGF or vehicle control and time-lapse movies were recorded over the course of 4 hr. The number of extending axon branches (**D**) was quantified over the course of the movies and the net growth rate of axon branch was calculated (**E**). Data represents mean ± SEM from 39 neurons, n = 3 independent experiments. ***p<0.001; ****p<0.0001; unpaired Student's ttest. (**F**) Hippocampal neurons were transfected with mCherry-UtrCH and imaged in TIRF-mode at 3 DIV. Using this approach, actin patches, membrane bending protrusions and filopodia could be identified and differentiated. Time-lapse movies were recorded over the course of 2 min (actin patches) or 10 min (protrusions and filopodia) to study the dynamic formation of such events. Scale bar 2 µm. (**G**) The number of newly forming actin patches during 2 min per 10 µm axon segment were counted before and after VEGF stimulation (100 ng/ml). Data represents mean ± SEM from at least 11 neurons of 2 independent experiments. n.s. not significant; paired Student's ttest. (**H,I**) The number and the size of newly forming protrusions and filopodia were analyzed during the course of 10 min before and after VEGF stimulation (100 ng/ml). Data represents mean ± SEM from at least 12 neurons of at least three independent experiments. n.s. not significant; paired Student's ttest.

The online version of this article includes the following video, source data, and figure supplement(s) for figure 3:

**Source data 1.** Raw data and statistical analysis of graphs of Figure 3.

**Figure supplement 1.** VEGF stimulation does not affect neurite growth and axon branch retraction.

**Figure supplement 1—source data 1.** Raw data and statistical analysis of graphs of Figure 3—figure supplement 1.

**Figure 3—video 1.** Hippocampal axon branching - unstimulated.

*Figure 3 continued on next page*

*Figure 3 continued*

https://elifesciences.org/articles/49818#fig3video1

**Figure 3—video 2.** Hippocampal axon branching – VEGF stimulated.

https://elifesciences.org/articles/49818#fig3video2

**Figure 3—video 3.** Actin-dynamics in axons of hippocampal neurons.

https://elifesciences.org/articles/49818#fig3video3

**Figure 3—video 4.** Actin patch-to-filopodia transition in axons of hippocampal neurons.

https://elifesciences.org/articles/49818#fig3video4

DIV hippocampal neurons resulted in increased phosphorylation of VEGFR2 (Y1175; *Figure 4—figure supplement 1A*).

To better understand the mechanism via which VEGFR2 signaling controls axon branching, we investigated the specific localization of VEGFR2 along the axon and its position or motility upon VEGF stimulation. For this, we transfected 1 DIV hippocampal neurons with a plasmid encoding a GFP-tagged version of the human VEGFR2 (VEGFR2-GFP). Confocal microscopy analysis of hippocampal neurons at 3 DIV revealed localization of the recombinant receptor in the axonal growth cones, branching points and actin-rich regions (*Figure 4—figure supplement 1B–1D*). To further investigate VEGFR2 localization and dynamics as well as its correlation with actin protrusions and filopodia, we co-transfected hippocampal neurons with the VEGFR2-GFP and the mCherry-UtrCH plasmids. Analysis of double transfected neurons confirmed that VEGFR2 localizes to actin-rich membrane protrusions and filopodia (*Figure 4A*). To analyze the effect of VEGF on the motility and directed localization of VEGFR2, we stimulated hippocampal neurons with VEGF for 5 min and performed live imaging along the axon over the course of 60 s before and after VEGF stimulation. Analysis of VEGFR2-GFP kymographs (*Figure 4—figure supplement 1E*) showed that stimulation with VEGF increased the motility of VEGFR2 in the axonal segments of the hippocampal neurons (*Figure 4B and C*; *Figure 4—video 1* and *Figure 4—video 2*). VEGF stimulation led to an enhanced movement of VEGFR2-GFP towards (converging) membrane protrusions and filopodia (*Figure 4D*). Consistently, blockage of the receptor using an anti-VEGFR2 functional blocking antibody abolished this motility (*Figure 4—figure supplement 1F–1H*; *Figure 4—video 3* and *Figure 4—video 4*).

Together, these data show that VEGFR2 moves towards actin-rich structures upon VEGF stimulation and suggest that VEGFR2 might subsequently locally stabilize the engaged actin structure to form a new emerging axon branch.

In cerebellar granule cells and commissural neurons, where VEGFR2 is required for migration and axon guidance respectively, VEGF/VEGFR2-mediated effects occur via SFKs activation (*Bellon et al., 2010*; *Chauvet et al., 2007*; *Meissirel et al., 2011*; *Ruiz de Almodovar et al., 2011*). Moreover, SFKs can remodel the actin cytoskeleton (*Brunton et al., 2004*; *Wang et al., 2011*; *Winograd-Katz et al., 2011*). Thus, we next investigated whether VEGF-induced axon branching in hippocampal neurons may also require the activation of SFKs. To address this, we first determined the levels of SFKs activation upon VEGF stimulation of 1 DIV neurons. We immunostained control and VEGF treated hippocampal neurons with a specific antibody that detects the activated form of SFKs. Quantitative analysis of the fluorescence intensity showed an increased activation of SFKs at axonal growth cones (GC) as well as in the primary axon upon treatment with VEGF (*Figure 4E*).

We next explored whether SFKs activity is required for the VEGF-induced morphological changes in hippocampal neurons. To this end, we used 1 µM PP2 (a widely used SFK inhibitor) to block SFKs activity in cultured hippocampal neurons or its inactive analog, PP3, as a control. Analysis of the axon branch number and length revealed a significant increase in PP3-treated control neurons upon VEGF stimulation, which was completely abolished upon PP2 treatment (*Figure 4F and G*). The requirement of SFKs activity was further confirmed by monitoring branching dynamics using time-lapse video microscopy in 1 DIV hippocampal neurons. Blocking SFKs completely abrogated the VEGF-mediated increase in the number of axon branches and the branch growth rate (*Figure 4H and I*).

Thus, VEGF can induce axon branching via both the mobilization of VEGFR2 to actin-rich structures and the activation of SFKs.

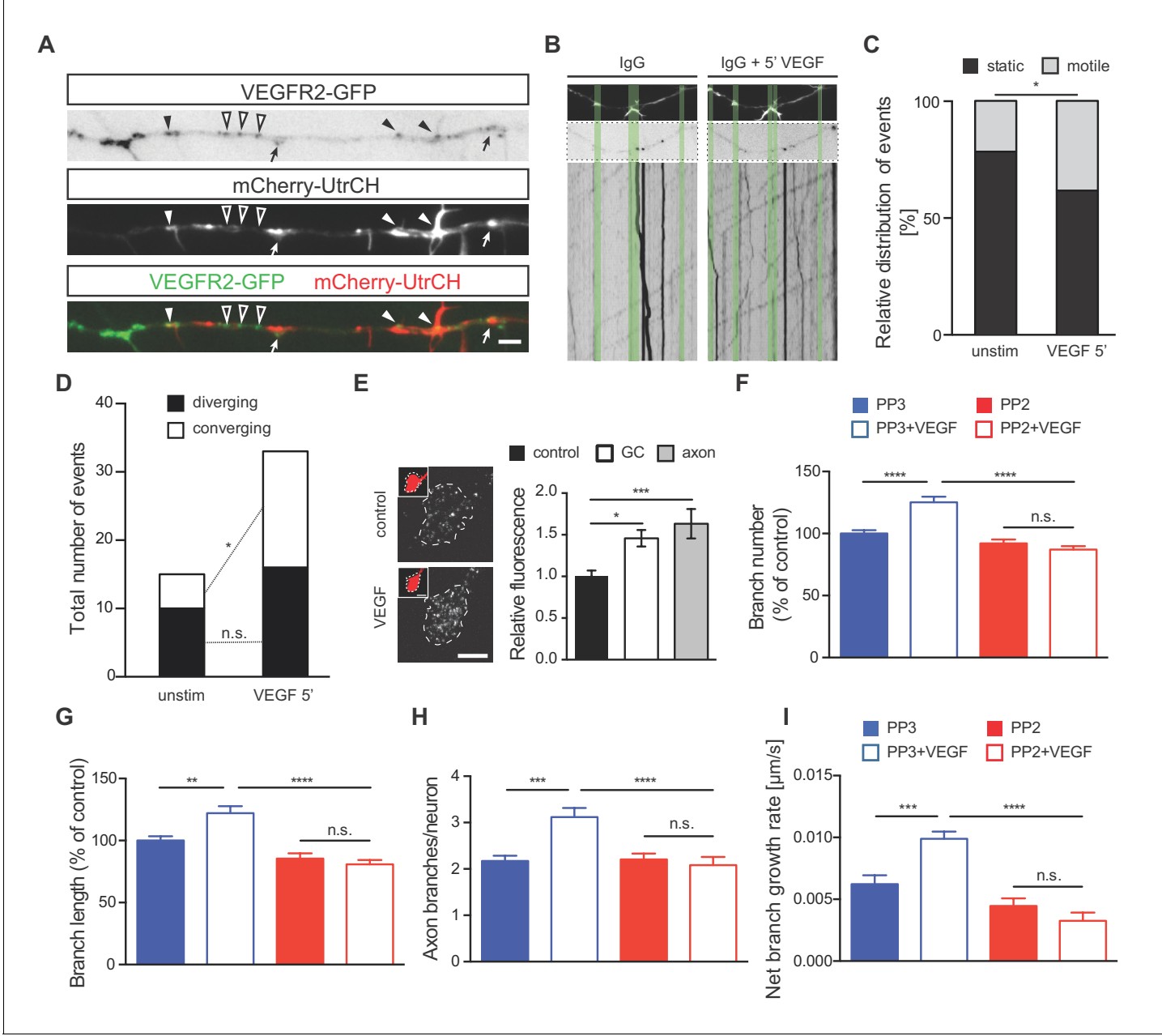

**Figure 4.** VEGF stimulation induces VEGFR2 motility towards actin-rich protrusions and filopodia and promotes axonal branching in a Src-dependent manner. (**A**) Hippocampal neurons were double transfected with mCherry-UtrCH and VEGFR2-GFP plasmids. TIRF microscopy reveals the localization of VEGFR2-GFP punctae at actin nucleation sites (arrows), the base of membrane protrusions and filopodia (black arrowheads) and along axon segments with neither actin nucleation nor protrusion sites (open arrowheads). Scale bar 5 μm. (**B–D**) To analyze relative VEGFR2-GFP motility and directionality, kymographs of VEGFR2-GFP mobility were generated from time-lapse movies of VEGFR2-GFP and mCherry-UtrCH co-transfected hippocampal neurons (note that IgG treatment is present in these neurons as the movies and kymographs were performed at the same time as the movies and resultant kymographs shown in **Figure 4—figure supplement 1 F-H**, and thus they serve as the corresponding controls) Upper images shows mCherry-UtrCH status at t = 0 s, middle images shows VEGFR2-GFP at t = 0 s, lower panels shows the kymograph of VEGFR2-GFP over the course of 1 min (**B**). Relative VEGFR2-GFP motility at protrusions and filopodia (**C**) as well as the direction of VEGFR2-GFP movement (**D**) were analyzed over the course of 1 min before and 5 min after VEGF stimulation (100 ng/ml). Per condition,>13 neurons were analyzed of at least three independent experiments. n.s. not significant, *p<0.05; Chi-squared test. (**E**) Hippocampal neurons were stimulated with or without VEGF and fixed after 5 min. Immunostaining for p-Src was performed and the relative fluorescence was analyzed in the axonal growth cone (GC) or along the axon. Data are represented as mean ± SEM, from at least three independent experiments. *p<0.05; ***p<0.001; One-way ANOVA. (**F,G**) 1 DIV hippocampal neurons were treated with 1 μM PP2 (a widely used SFK inhibitor) or PP3 (control) for 1 hr prior stimulation with or without 100 ng/ml VEGF for 48 hr. Quantifications of axonal branch number (**F**) and branch length (**G**) are shown. Data are represented as % of non-stimulated control. Mean ± SEM of at least three independent experiments. n.s.

*Figure 4 continued on next page*

*Figure 4 continued*

not significant; **p<0.01; ****p<0.0001; Two-way ANOVA. (H,I) 1 DIV hippocampal neurons were pretreated with 1 μM PP2 or PP3 for 1 hr. After stimulation with 50 ng/ml VEGF or vehicle control time-lapse movies were recorded over the course of 4 hr. The number of extending axon branches is quantified over the course of the movies (H) and the net growth rate of axon branch was calculated (I). Mean ± SEM of at least three independent experiments. n.s. not significant; ***p<0.001; ****p<0.0001; Two-way ANOVA.

The online version of this article includes the following video, source data, and figure supplement(s) for figure 4:

**Source data 1.** Raw data and statistical analysis of graphs of Figure 4.

**Figure supplement 1.** VEGFR2 localizes at growth cones, axon branch points and actin nucleation sites of hippocampal neurons and is activated upon VEGF stimulation.

**Figure supplement 1—source data 1.** Raw data and statistical analysis of graphs of Figure 4—figure supplement 1.

**Figure 4—video 1.** VEGF-GFP motility in hippocampal axons – IgG unstimulated.

https://elifesciences.org/articles/49818#fig4video1

**Figure 4—video 2.** VEGF-GFP motility in hippocampal axons – IgG + 5′ VEGF.

https://elifesciences.org/articles/49818#fig4video2

**Figure 4—video 3.** VEGF-GFP motility in hippocampal axons – αVEGFR2 unstimulated.

https://elifesciences.org/articles/49818#fig4video3

**Figure 4—video 4.** VEGF-GFP motility in hippocampal axons – αVEGFR2 + 5′ VEGF.

https://elifesciences.org/articles/49818#fig4video4

## Nervous system specific VEGFR2 knockout mice show increased axon branching and deficits in synapse development in vivo

To investigate whether VEGFR2 is crucial for the development of hippocampal neurons, we genetically deleted the gene encoding VEGFR2 (*Kdr*) in the nervous system by crossing *Kdr*^lox/- mice (*Haigh et al., 2003*) with the *Nestin-cre* mouse line (*Tronche et al., 1999*). PCR analysis and Western blotting of lysates from isolated hippocampal neurons confirmed the absence of *VEGFR2* mRNA and protein in hippocampal neurons of *Nestin-cre^+;Kdr*^lox/- animals compared to *Nestin-cre^-;Kdr*^lox/- littermates (from hereon termed *Nes-cre;Kdr*^lox/- and control, respectively) (see accompanying manuscript *Harde et al., 2019*) We first characterized the cytoarchitecture of the entire hippocampus in *Nes-cre;Kdr*^lox/- mice. We labeled brain sections of P10 *Nes-cre;Kdr*^lox/- and control mice with TO-PRO-3 (labels all nuclei), NeuN (labels neuronal nuclei) and L1 (labels axonal tracks) and analyzed the gross morphology of the hippocampus, the number of hippocampal neurons, the size of the CA3 area and the axonal tracks. Morphometric analysis revealed that hippocampal anatomy was not affected in *Nes-Cre;Kdr*^ox/- mice compared to control littermates (*Figure 5—figure supplement 1A–1E*), consistent with what was previously described in *Nes-cre;Kdr*^lox/lox adult mice (*Hermann et al., 2013*).

To characterize the hippocampus in *Nes-cre;Kdr*^lox/- mice in detail, we analyzed the morphology of CA3 hippocampal neurons at a high resolution level by focusing on their axons and investigated whether VEGFR2 plays a role in axon branching of these neurons in vivo. We set up a technique to label single axons of CA3 pyramidal neurons in situ as it has been previously described for DRG neurons (*Schmidt and Rathjen, 2011*). We used horizontal thick vibratome sections of postnatal brains and placed small DiI crystals in the *stratum pyramidale* of the CA3 region (*Figure 5—figure supplement 2A and B*). Short incubation with the DiI crystals allowed the labeling of a small number of axons and the identification of single axon segments and their branches (*Figure 5—figure supplement 2C–2E*). Labeled CA3 axons were imaged and traced in the *stratum radiatum* of the CA1 region where they project to (*Figure 5—figure supplement 2C–2E*). To validate this approach, we first characterized axon branching of CA3 neurons in different postnatal developmental stages of wildtype mice. Consistent with previous published data (*Gomez-Di Cesare et al., 1997*), we observed a steady increase in the number of axon branches until P16, which subsequently declined and reduced by P30 (*Figure 5—figure supplement 2F*). Next, we determined whether axon branching was affected in the brains of *Nes-cre;Kdr*^lox/- mice at P10. Our data revealed that in the absence of VEGFR2 there is a significant increase of axon branches of CA3 pyramidal neurons (*Figure 5A and B*). Notably, these branches were shorter compared to control littermates (*Figure 5C*).

To further examine whether the increase in the number of axon branches would correlate with an increase in the number of synaptic connections, we quantified the number of functional synapses in the *stratum radiatum* of the CA1 area using transmission electron microscopy. Synapses were

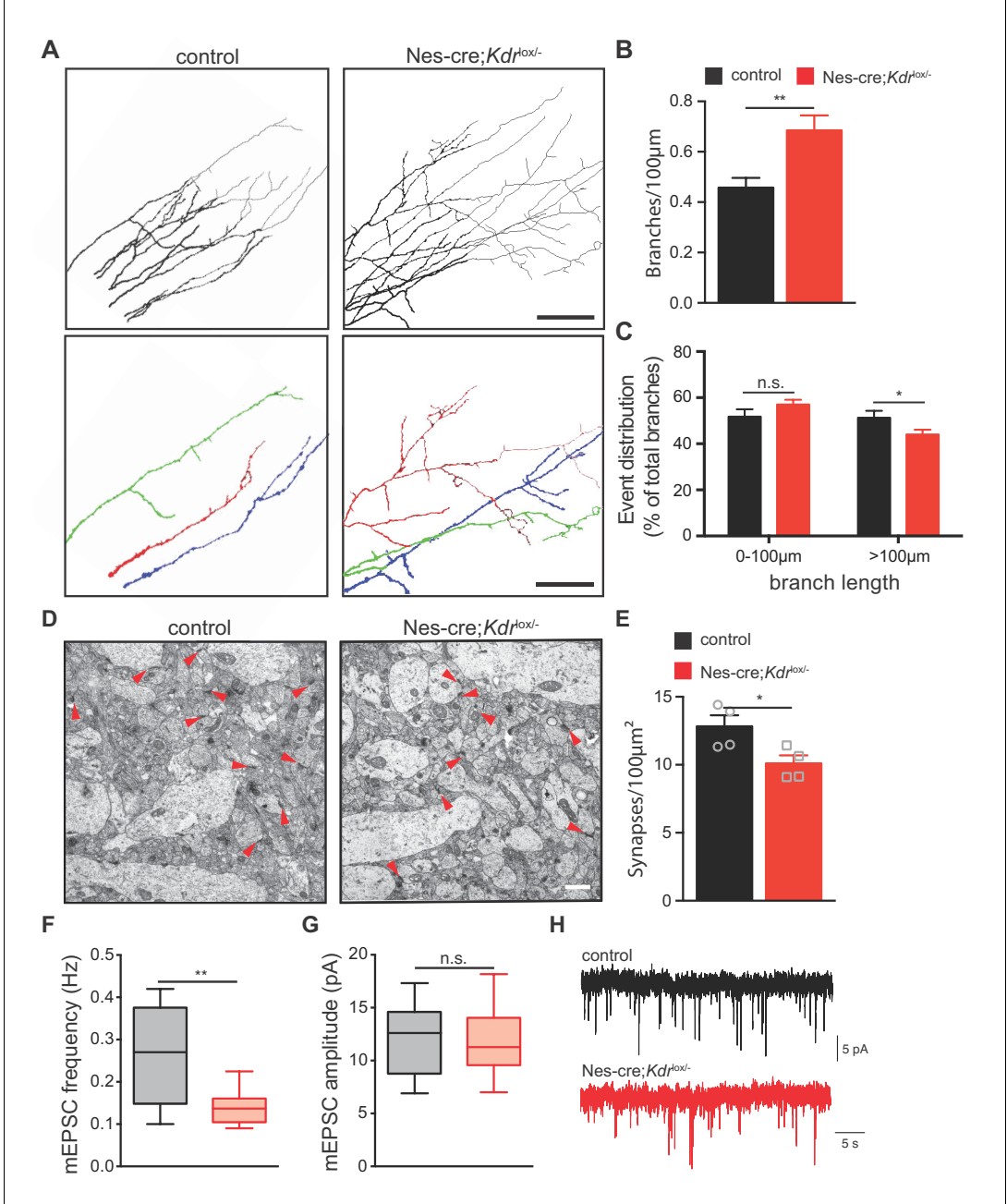

**Figure 5.** Nes-cre;Kdr^lox/-^mice show defects in axon branching and synapse density in vivo. (**A**) Representative tracings in the CA1 stratum radiatum of CA3 DiI labeled axons from P10 control and Nes-cre;*Kdr*^lox/-^ mice. Selected individual axon traces are shown highlighted in different colors in the panels below. Scale bar 100 μm. (**B,C**) Quantification of the axon branch number per 100 μm axon length (**B**) as well as their length distribution (**C**). Data are represented as mean ± SEM from n = 15 control and n = 18 Nes-cre;*Kdr*^lox/-^ pups of 6 independent litters (**B,C**). n.s. not significant; *p<0.05; **p<0.01; unpaired Student's ttest. (**D**) Representative electron micrographs of the hippocampal CA1 stratum radiatum from P10 control and Nes-cre;*Kdr*^lox/-^ mice. Red arrowheads indicate synapses. Scale bar 1 μm. (**E**) Quantification of synapse density in control and Nes-cre;*Kdr*^lox/-^ mice. Synapse numbers from 8 fields of view of n = 4 mice from each genotype were analyzed. *p<0.05; unpaired Student's ttest. (**F,G**) Average mEPSC frequency (**F**) and amplitude (**G**) in control and Nes-cre;*Kdr*^lox/-^ mice. n.s. not significant; **p<0.01; unpaired Student's ttest. (**H**) Representative mEPSC recordings in control and Nes-cre;*Kdr*^lox/-^ mice.

The online version of this article includes the following source data and figure supplement(s) for figure 5:

**Source data 1.** Raw data and statistical analysis of graphs of Figure 5.

**Figure supplement 1.** *Nes-cre;Kdr*^lox/-^ mice do not show defects in the overall hippocampal morphology and cytoarchitecture.

**Figure supplement 1—source data 1.** Raw data and statistical analysis of graphs of Figure 5—figure supplement 1.

*Figure 5 continued on next page*

**Figure supplement 2.** Experimental setup for DiI labeling of CA3 axons in ex vivo slices.
**Figure supplement 3.** VEGF/VEGFR2-mediated regulation of axon branching does not require Nrp1.
**Figure supplement 3—source data 1.** Raw data and statistical analysis of graphs of Figure 5—figure supplement 3.
**Figure supplement 4.** VEGF/VEGFR2-mediated regulation of axon branching requires active endocytosis but not EphrinB2.
**Figure supplement 4—source data 1.** Raw data and statistical analysis of graphs of Figure 5—figure supplement 4.

identified based on the presence of the synaptic cleft, postsynaptic densities and presynaptic vesicles. We observed that the number of synapses was decreased in *Nes-cre;Kdr*$^{lox/-}$ mice when compared to their control littermates (*Figure 5D and E*). Consistently, whole-cell patch clamp recordings in CA1 neurons of *Nes-cre;Kdr*$^{lox/-}$ mice revealed a decrease in the frequency of miniature excitatory postsynaptic currents (mEPSCs) (*Figure 5F and H*). The amplitude of mEPSCs was unchanged, suggesting that the strength of newly formed CA3-CA1 synapses is similar in both control and *Nes-cre;Kdr*$^{lox/-}$ mice (*Figure 5G*).

Together, these results indicate that VEGFR2 is required for proper axon branching and synapse formation in CA3 pyramidal neurons. They further show that *Nes-cre;Kdr*$^{lox/-}$ mice develop an increased amount of axon branches that remain immature and do not form functional synapses within the CA1 *stratum radiatum*, suggesting that perhaps this increase in branching is built as a compensatory response.

## Absence of VEGFR2 in hippocampal neurons results in newly formed axon branches due to increased filopodia number and growth rate

To better understand the mechanisms underlying the increased axon branching upon loss of VEGFR2, we isolated hippocampal neurons from *Nes-cre;Kdr*$^{lox/-}$ and control mice and analyzed their axonal morphology in vitro. Morphometric analysis of knockout neurons at 3 DIV revealed a significant increase in the number and length of axon branches (*Figure 6A–6C*). These results are in agreement with the axonal phenotype we observed in vivo in *Nes-cre;Kdr*$^{lox/-}$ animals and confirm that VEGFR2 is important for proper axonal development. To further explore whether this phenotype results from increased axon branch elongation or reduced retraction, we performed wide field time-lapse movies. We followed the dynamic branching process in hippocampal neurons from *Nes-cre;Kdr*$^{lox/-}$ mice or control neurons over the course of 4 hr. Quantitative analysis showed that loss of VEGFR2 led to an increase of the net growth and net growth rate of newly forming branches without affecting the number and rate of axonal branch retraction (*Figure 6D and E*; *Figure 6—figure supplement 1A and B*).

Next, we determined the changes that occur at the level of actin patches and filopodia by analyzing mCherry-UtrCH transfected control and *Nes-cre;Kdr*$^{lox/-}$ hippocampal neurons. Using 2 min time-lapse TIRF movies we analyzed the dynamics of actin patch formation in hippocampal axons from *Nes-cre;Kdr*$^{lox/-}$ or control mice. We did not observe any changes in the formation of actin patches (*Figure 6F*). However, neurons from *Nes-cre;Kdr*$^{lox/-}$ mice showed a significantly increased percentage of filopodia formed in 10 min, as well as increased percentage of events larger than 2 μm (*Figure 6G and H*). Consistently with a functional signaling role of VEGFR2 in axon branching, similar results were obtained by blocking VEGFR2 signaling with an anti-VEGFR2 functional blocking antibody (α-VEGFR2) in WT isolated neurons (*Figure 6—figure supplement 1A and C–K*).

Our in vitro analysis shows that lack of VEGFR2 activation correlates with enhanced filopodia formation and increased net growth rate of axon branches, which together result in increased number of axon branches. Taken together, these in vitro results are consistent with the in vivo phenotype of *Nes-cre;Kdr*$^{lox/-}$ mice. In vivo, the increased axon branching is non-functional as axon branches fail to form synaptic contacts with the CA1 and present impaired functionality.

## VEGF-mediated axon branching via VEGFR2 does not require Neuropilin 1 nor EphrinB2

Neuropilin 1 (Nrp1) is expressed in hippocampal neurons during development (*Figure 5—figure supplement 3A*; *He and Tessier-Lavigne, 1997*). As VEGF also binds Nrp1 and can induce signaling dependent or independent of VEGFR2 (*Roth et al., 2016*), we questioned whether Nrp1 is required

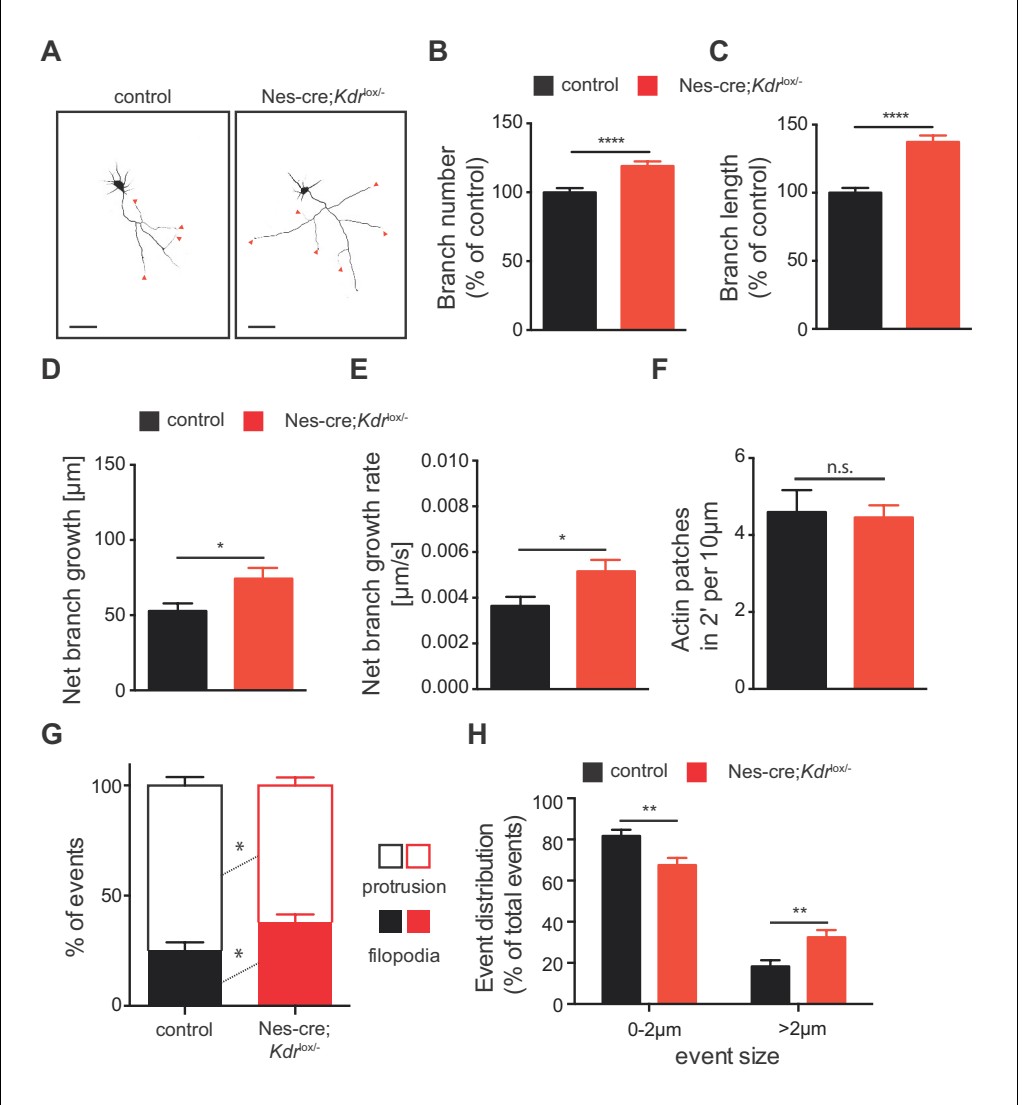

**Figure 6.** VEGFR2 deficiency in hippocampal neurons promotes axonal branching by increasing filopodia formation. (**A**) Representative images of 3 DIV hippocampal neurons isolated from control and *Nes-cre;Kdr*<sup>lox/-</sup> embryos and stained beta-III tubulin. Axon branches are indicated by red arrowheads. Scale bars 50 μm. (**B,C**) Quantification of the axon branch number (**B**) and branch length (**C**) of control and *Nes-cre;Kdr*<sup>lox/-</sup> neurons at 3 DIV. Data are represented as % of control. Mean ± SEM,>120 neurons from n = 6. ****p<0.0001; unpaired Student's ttest. (**D,E**) Time-lapse movies over the course of 4 hr were recorded from 1 DIV hippocampal neurons of control and *Nes-cre;Kdr*<sup>lox/-</sup> mice. The net axon branch growth was quantified over the course of the movies (**D**) and the growth rate of axon branch was calculated (**E**). Data represents mean ± SEM of at least three independent experiments. *p<0.05; unpaired Student's ttest. (**F**) The number of newly forming actin patches during 2 min per 10 μm axon segment was counted in neurons of control and *Nes-cre;Kdr*<sup>lox/-</sup> mice. Data represents mean ± SEM of at least three independent experiments. n.s. not significant; unpaired Student's ttest. (**G,H**) The number (**G**) and the size (**H**) of newly formed protrusions and filopodia was analyzed during the course of 10 min in neurons of control and *Nes-cre;Kdr*<sup>lox/-</sup> mice. Data represents mean ± SEM of at least three independent experiments. *p<0.05; **p<0.01; unpaired Student's ttest.

The online version of this article includes the following source data and figure supplement(s) for figure 6:

**Source data 1.** Raw data and statistical analysis of graphs of Figure 6.

**Figure supplement 1.** α-VEGFR2 antibody treatment recapitulates the phenotype observed in Nes-cre;*Kdr*<sup>lox/-</sup> mice.

**Figure supplement 1—source data 1.** Raw data and statistical analysis of graphs of Figure 6—figure supplement 1.

for VEGF-induced axon branching. For this, we blocked Nrp1 in hippocampal neurons using an anti-Nrp1 neutralizing antibody (α-Nrp1) and analyzed axon branching. In the presence of α-Nrp1 VEGF was still able to induce axon branching (*Figure 5—figure supplement 3B and C*), thus indicating that Nrp1 is dispensable for VEGF-mediated effect on branching. Moreover, blockage of Nrp1 in the presence of α-VEGFR2 did not normalize the increased axon branching (*Figure 5—figure supplement 3D and E*), suggesting that in the absence of VEGFR2 Nrp1 is not a receptor via which VEGF could alternatively signal to regulate axon branching.

In the accompanying manuscript (*Harde et al., 2019*), we show that ephrinB2 is expressed in hippocampal neurons during development and that its expression is required for VEGFR2 internalization and VEGF-induced dendritic branching. We therefore tested whether VEGFR2 internalization is required for VEGF-induced axon branching and whether ephrinB2 would also play a role in regulating axon branching. Inhibition of VEGFR2 internalization with dynasore prevented the increased axon branching upon VEGF stimulation (*Figure 5—figure supplement 4A and B*), indicating that VEGFR2 internalization is indeed required for the response of the axon to VEGF. However, stimulation of primary ephrinB2 knockout (*Efnb2$^{-/-}$*) hippocampal neurons with VEGF resulted in a similar increase in axon branching as control neurons (*Figure 5—figure supplement 4C and D*), suggesting that ephrinB2 is not regulating VEGFR2 function for its effects in axon branching. Consistently, analysis of CA3 hippocampal axon branching in vivo in compound mutant mice heterozygous for *ephrinB2* and *Kdr* in neurons (*Kdr-ephrinB2* compound mice; *Nes-cre;Kdr$^{lox/+}$;Efnb2$^{lox/+}$*; see accompanying manuscript) using the DiI labeling approach as described before did not show any difference when compared to control littermates (*Figure 5—figure supplement 4E*).

Altogether, these results show that VEGF/VEGFR2 regulation of hippocampal axon branching does not depend on other co-receptors such as Nrp1 and EphrinB2. They also highlight that the molecular mechanism of VEGF/VEGFR2 signaling, regulating hippocampal axon branching or hippocampal dendritogenesis and spine morphogenesis, are distinctly different.

## Discussion

This study reports a novel molecular mechanism by which direct VEGF/VEGFR2 signaling on hippocampal pyramidal neurons regulates axon branching during development.

Our expression analysis shows that VEGF and VEGFR2 are regionally, dynamically and temporally expressed in the developing hippocampus at the time when axon arborization takes place. VEGF is expressed in CA neurons, glia and endothelial cells and its expression in neurons decreases during the first postnatal weeks. During late embryonic and early postnatal (first two weeks) hippocampus development, neuronal expression of VEGFR2 is restricted to CA3 neurons. These expression patterns are consistent with later postnatal and adult stages where VEGF expression becomes restricted to adult astrocytes and VEGFR2 is primarily found in vessels (*Licht et al., 2011*) and at the postsynaptic side of CA1 neurons (*De Rossi et al., 2016*).

We demonstrate that in hippocampal neurons VEGF activates VEGFR2 and SFKs to induce axon branching. While Nrp1 is also expressed in hippocampal neurons, its blockage does not inhibit VEGF-mediated increase in axon branching. Previous reports show that VEGF regulates cerebellar granule cell migration or commissural axon guidance in a SFK-dependent manner (*Meissirel et al., 2011*; *Ruiz de Almodovar et al., 2011*). Here we show that VEGF-induced axon branching requires VEGFR2 internalization and SFK activation. We found that VEGF stimulation induces activation of SFKs at growth cones and along the axon shaft. Thus, SFK activation seems to be a crucial step in VEGF-mediated effects in neurons. Upon VEGF stimulation, VEGFR2 motility and migration towards actin-rich protrusions and filopodia is enhanced, suggesting that localization of VEGFR2 in those cellular structures contributes to the final outcome of increased axon branching.

Interestingly, we observed that both stimulation with VEGF as well as the inactivation of VEGFR2 result in an increased number of axon branches (*Figure 7*). In all cases we see that branch growth is enhanced and that there are no differences in branch retraction. How can the activation and the loss of the same signaling receptor lead to a similar phenotype? We show that although the phenotypic outcome is the same, the cause for the increase in axon branch number differs between the activation and the loss of VEGFR2. In the case of VEGFR2 activation we find that VEGF stimulation does not change the percentage of forming filopodia but it does increase the number of branches (*Figure 7*). VEGF stimulation further increases the growth rate of axon branches. We propose, that

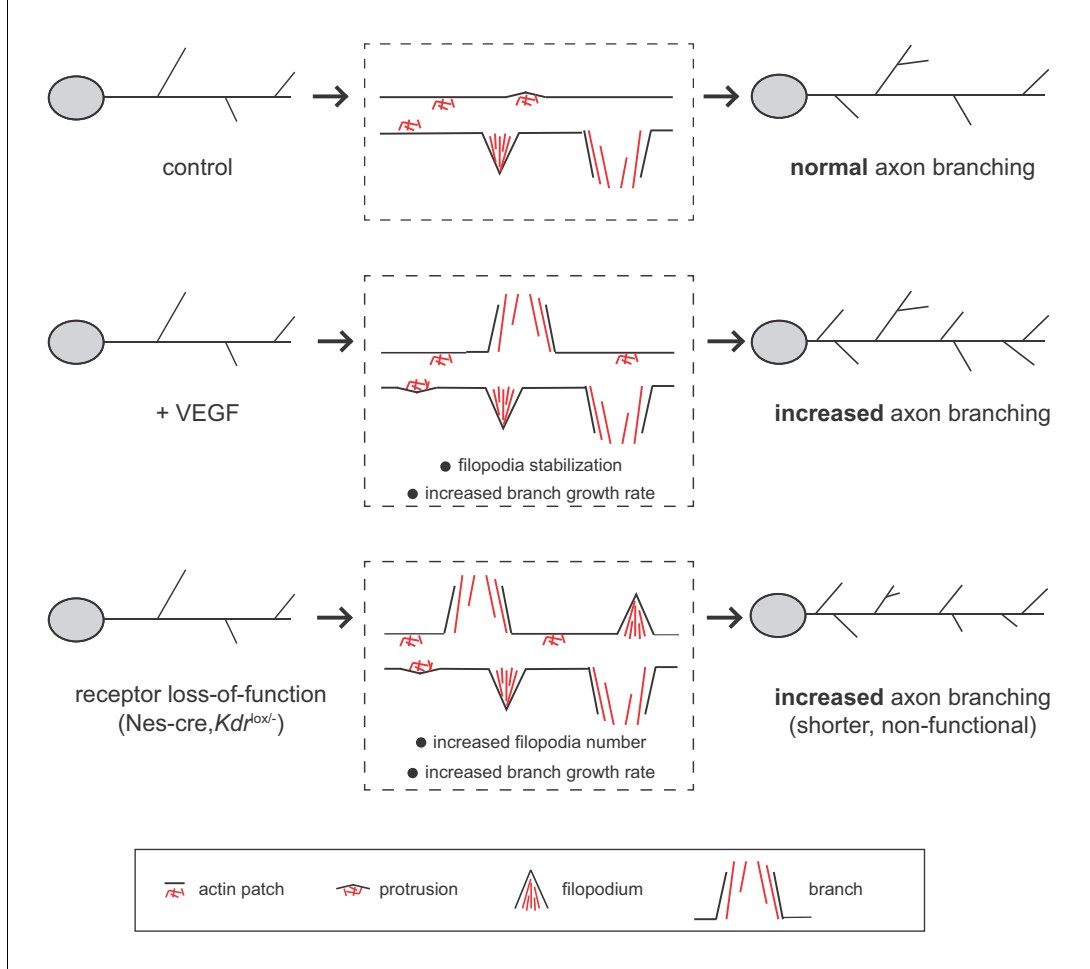

**Figure 7.** Working model for VEGF/VEGFR2 function in hippocampal axon branching. VEGF stimulation induces axon branching by promoting the growth of newly formed branches without affecting filopodia formation. Loss-of-functional VEGFR2 also results in increased axon branching, but in this case from an increased number of filopodia and enhanced growth rate. However, in the loss of function situation, the generated branches are shorter and non-functional.

VEGF induces axon branching by stabilizing and promoting the growth of branches once the filopodia have already been formed. In contrast, upon VEGFR2 loss, the percentage of forming filopodia increases, as well as the growth rate, overall resulting in an increased number of axon branches (*Figure 7*). One hypothesis could be that in absence of the receptor the axon branch cannot be properly stabilized and grow, and thus the axon branches that are formed remain shorter.

Similarly, disruption of VEGFR2 signaling in vivo (in *Nes-cre;Kdr*lox/- mice) results in axonal hyperbranching of CA3 hippocampal neurons. Despite the increase in axon branch number, we observed a reduced number of synapses in the CA1 area, indicating that the branches are not functional and rather immature. An increase in the number of immature non-functional axon branches has been described in DCN neurons in *Drosophila* upon inhibition of EGFR signaling (*Zschätzsch et al., 2014*). In our case, the overshooting of branches could occur as a compensatory response with the aim to restore proper axon branching.

The accompanying manuscript by *Harde et al. (2019)* shows that the VEGF/VEGFR2 signaling effect is not limited to the axon, but this ligand/receptor pair also regulates dendritic development and spine morphogenesis in hippocampal neurons. While in both compartments the internalization of VEGFR2 is required for VEGF signal transduction, the presence of EphrinB2 and its regulatory function seems to specifically contribute to dendritic development and to be dispensable for axonal branching. In line with this, we find that inactivation of VEGFR2 has an opposite effect in the axon and in the dendrites. While absence of VEGFR2 results in increased branching of hippocampal

axons, it leads to a less branched dendritic network. A differential effect in dendrites and axons has been also described for other signaling molecules such as Sema3A, which promotes dendritic growth but restricts axon growth in cultured hippocampal neurons, or the leucine zipper kinase (DLK) pathway, which restrains dendritic growth but promotes axon growth (*Wang et al., 2013*; *Wang et al., 2014*). Therefore, VEGF/VEGFR2 signaling falls into the category of bimodal regulators that differentially direct axonal and dendritic development in an opposite manner.

A study in *Drosophila* elegantly showed that intrinsic asymmetry in EGFR localization and local activation of EGFR signaling within filopodia is required for proper axon branching of dorsal cluster neurons and that both EGFR activation and loss of function results in increase axon branching due to a defect in pruning (*Zschätzsch et al., 2014*). Our data provide for the first time evidence that in mammals similar processes can also modulate axon branching. In particular, we find that appropriate levels of VEGFR2 signaling in hippocampal neurons are required for proper axon branching by a mechanism that involves branch growth but not branch pruning. We show that upon VEGF stimulation and activation of VEGFR2 signaling, VEGFR2 motility increases within actin protrusions and filopodia, correlating with an increase in branch number. Two possibilities, non-exclusive, could explain the increased axon branching upon VEGFR2 deletion. On the one hand, the lack of VEGFR2 could lead to a decrease in the dynamics of protrusions turnover and thus increase the probability of a protrusion to become a filopodium. Our data support such a model as we observe that the absence of VEGFR2 increases the percentage of filopodia. On the other hand, a compensatory mechanism, yet unidentified, might become activated to overcome the inhibition of VEGFR2, resulting in higher branch number than in control conditions.

# Materials and methods

## Key resources table

| Reagent type (species) or resource | Designation | Source or reference | Identifier | Additional information |
|---|---|---|---|---|
| Genetic reagent (*Mus musculus*) | $Kdr^{lox/lox}$ | *Haigh et al., 2003* | | |
| Genetic reagent (*Mus musculus*) | Nestin-cre | *Tronche et al., 1999* | | |
| Genetic reagent (*Mus musculus*) | Vegfr2-GFP | *Ema et al., 2006* | | |
| Genetic reagent (*Mus musculus*) | $Efnb2^{lox/lox}$ | *Grunwald et al., 2004* | | |
| Antibody | rabbit anti-GFP polyclonal antibody | Invitrogen | Cat# A11122 RRID:AB_221569 | 1:200 IF |
| Antibody | chicken anti-GFP polyclonal antibody | Aves | Cat# GFP-1020 RRID:AB_10000240 | 1:200 IF |
| Antibody | rabbit anti-GFAP polyclonal antibody | Dako | Cat# Z0334 RRID:AB_10013382 | 1:100 IF |
| Antibody | mouse anti-NeuN monoclonal antibody | Millipore | Cat# MAB377 RRID:AB_2298772 | 1:50 IF |
| Antibody | rat anti-L1 monoclonal antibody | Millipore | Cat# MAB5272 RRID:AB_2133200 | 1:100 IF |
| Antibody | rabbit anti-Calretinin polyclonal antibody | Millipore | Cat# MAB5054 | 1:500 IF |
| Antibody | goat anti-Nrp1 polyclonal antibody | R and D | Cat# AF566 RRID:AB_355445 | 2 µg/ml for functional blocking |
| Antibody | goat anti-VEGFR2 polyclonal antibody | R and D | Cat# AF644 RRID:AB_355500 | 1:15 IF |
| Antibody | rabbit anti-VEGFR2 monoclonal antibody | Cell Signaling | Cat# 2479 RRID:AB_2212507 | 1:1000 WB |

*Continued on next page*

*Continued*

| Reagent type (species) or resource | Designation | Source or reference | Identifier | Additional information |
|---|---|---|---|---|
| Antibody | rabbit anti-(phospho) VEGFR2 (Y1175) | Cell Signaling | Cat# 2478 RRID:AB_331377 | 1:500 WB |
| Antibody | rabbit anti-(phospho)SFK (Y416) monoclonal antibody | Invitrogen | Cat# 44660G RRID:AB_2533714 | 1:200 IF |
| Antibody | mouse anti-beta-III-tubulin monoclonal antibody | Sigma | Cat# T5076 RRID:AB_532291 | 1:150 IF |
| Antibody | goat anti-VE-Cadherin polyclonal antibody | R and D | Cat# AF1002 RRID:AB_2077789 | 1:1000 WB |
| Antibody | donkey anti-rabbit Alexa488 polyclonal antibody | Jackson Immunoresearch | Cat# 711-545-152 RRID:AB_2313584 | 1:1000 IF |
| Antibody | donkey anti-rabbit Alexa568 polyclonal antibody | Life Technologies | Cat# A10042 RRID:AB_2534017 | 1:1000 IF |
| Antibody | donkey anti-goat Alexa568 polyclonal antibody | Molecular Probes | Cat# A11057 RRID:AB_142581 | 1:1000 IF |
| Antibody | goat anti-mouse Alexa568 polyclonal antibody | Invitrogen | Cat# A11031 RRID:AB_144696 | 1:1000 IF |
| Antibody | goat anti-rat Alexa 568 polyclonal antibody | Invitrogen | Cat# A11077 RRID:AB_2534121 | 1:1000 IF |
| Antibody | goat anti-chicken Alexa488 polyclonal antibody | Molecular Probes | Cat# A11039 RRID:AB_142924 | 1:1000 IF |
| Antibody | donkey anti-goat HRP polyclonal antibody | Jackson Immunoresearch | Cat# 705-035-147 RRID:AB_2313587 | 1:1000 IF |
| Antibody | donkey anti-rabbit HRP polyclonal antibody | Jackson Immunoresearch | Cat# 711-035-152 RRID:AB_10015282 | 1:1000 IF |
| Antibody | donkey anti-mouse HRP polyclonal antibody | Jackson Immunoresearch | Cat# 715-035-150 RRID:AB_2340770 | 1:1000 IF |
| Chemical compound, drug | Phalloidin-TRITC | Sigma | Cat# P1951 RRID:AB_2315148 | 1:400 IF |
| Chemical compound, drug | dynasore | Selleckchem | Cat# S8047 | 80 µM for functional blocking |
| Chemical compound, drug | DiI lipophilic tracer dye | Molecular Probes | Cat# D-3911 | |
| Chemical compound, drug | PP2 | Calbiochem | Cat# 529573 | 1 µM for functional blocking |
| Chemical compound, drug | PP3 | Calbiochem | Cat# 529574 | 1 µM for functional blocking |
| Recombinant DNA reagent | UtrCH-mCherry plasmid | *Burkel et al., 2007* | | |
| Recombinant DNA reagent | hVEGFR2-GFP fusion plasmid | | | |
| Commercial assay or kit | VEGF ELISA kit | R and D | Cat# MMV00 | |

## Animals

Nervous system-specific VEGFR2 conditional knockout mice (*Nes-cre;Kdr*$^{lox/-}$) were generated by crossing the previously described *Kdr*$^{lox/-}$ mice (*Haigh et al., 2003*) with Nestin-cre mice (*Tronche et al., 1999*). *Vegfr2*-GFP mice were obtained from Rüdiger Klein and previously described (*Ema et al., 2006*). *Kdr-EphrinB2* double-heterozygous conditional compound mice (*Nes-cre;*

*Kdr^{lox/+};EfnbB2^{lox/+}*) were generated by crossing mice double-homozygous floxed for *Kdr* and *ephrinB2* (*Kdr^{lox/lox};Efnb2^{lox/lox}*) to mice bearing one copy of the Nestin-cre transgene (*Nes-cre*) (see further accompanying manuscript by *Harde et al., 2019*). Single-heterozygous mice, where either only one *Kdr* allele (*Nes-cre;Kdr^{lox/+}*) or only one ephrinB2 allele (*Nes-cre;Efnb2^{lox/+}*) was deleted, were generated in a similar manner as the compound mice (see further accompanying manuscript by *Harde et al., 2019*). All mice were bred in the C57BL/6 background, except the *Vegrf2*-GFP mice, which were maintained in the ICR/HaL background. Wildtype C57BL/6NRJ mice were obtained from Janvier Laboratories. All animal experiments were conducted in accordance with the German institutional guidelines and ethical committees (references refs: FU1090, T60/12, T79/13, T49/15, T69/14, T46/16, T36/17, T48/18, and I19/13).

## Antibodies

Following primary antibodies were used for Immunofluorescence (IF) and Western blotting (WB): rabbit anti-GFP (1:200 IF, Invitrogen, A11122), chicken anti-GFP (1:200 IF, Aves, GFP-1020), rabbit anti-GFAP (1:100 IF, Dako, Z0334), mouse anti-NeuN (1:50 IF, Millipore, MAB377), rat anti-L1 (1:100 IF, Millipore, MAB5272), rabbit anti-Calretinin (1:500 IF, Millipore, MAB5054), goat anti-Nrp1 (2 µg/ml for function blocking, R and D, AF566), goat anti-VEGFR2 (1:15 IF, R and D, AF644), rabbit anti-VEGFR2 (1:1000 WB, Cell Signaling, 2479), rabbit anti-(phospho)VEGFR2 (Y1175) (1:500 WB, Cell Signaling, 2478), rabbit anti-(phospho/SFK (Y416) (1:200 IF, Invitrogen, 44660G), mouse anti-beta-III-tubulin (1:150 IF, Sigma, T5076), goat anti-VE-Cadherin (1:1000 WB, R and D, AF1002), Phalloidin (1:400 IF, Sigma, P1951). The following secondary antibodies were used: donkey anti-rabbit Alexa488 (Jackson Immunoresearch, 711-545-152), donkey anti-rabbit Alexa568 (Life Technologies, A10042), donkey anti-goat Alexa568 (Molecular Probes, A11057), goat anti-mouse Alexa568 (Invitrogen, A11031), goat anti-rat Alexa568 (Invitrogen, A11077), goat anti-chicken Alexa488 (Molecular Probes, A11039), donkey anti-goat HRP (Jackson Immunoresearch, 705–0350147), donkey anti-rabbit HRP (Jackson Immunoresearch, 711-035-152), donkey anti-mouse HRP (Jackson Immunoresearch, 715-035-150).

## In situ hybridization and immunohistochemistry

Wildtype and *Vegfr2*-GFP pups at the indicated ages received an anesthetic overdose by intraperitoneal injection of ketamine (180 mg/kg of body weight; Ketavet) and xylazine (10 mg/kg of body weight; Rompun) and were transcardially perfused first with PBS followed by 4% PFA. Brains were collected in 4% PFA and fixed overnight at 4°C. For cryosections, fixed and washed brains were placed in 30% sucrose and subsequently frozen in NEG-50 (Richard-Allan Scientific). For in situ hybridization, 20 µm cryosections were hybridized with DIG-labeled RNA probes for VEGFR2 (*Kdr*), VEGF (*Vegfa*) and Nrp1 mRNA as already described (*Ruiz de Almodovar et al., 2011*). For IF, cryosections were blocked for 1 hr at RT with PBS containing 1% BSA, 2% normal goat or normal donkey serum (Dianova) and 0.3% Triton X-100. Primary antibodies in blocking solution were added overnight at 4°C. Secondary antibodies in blocking solution were added for 2 hr at room temperature (RT), followed by nuclear staining with TO-PRO3 (1:1000, Life technologies). For combination with IF, ISH was performed first. Following postfixation, sections were washed and blocked for 4 hr at RT with PBS containing 1% BSA, 5% normal goat serum and 0.3% Triton. Subsequently, GFAP (1:100) or NeuN (1:50) antibodies were added in blocking solution overnight at 4°C. Corresponding secondary antibodies were incubated for 2 hr at room temperature followed by nuclear staining with TO-PRO3 (1:1000, Life technologies). Immunostainings were examined using a Zeiss confocal microscope (LSM510).

## Quantification of Phospho-SFK fluorescence

25.000 primary hippocampal neurons per coverslip were cultured for 24 hr and treated for 30 min with 80 µM dynasore or 0.1% DMSO. 5 min prior to stimulation with VEGF. At the same time, 1 mM sodium orthovanadate was added to the cultures. Neurons were stimulated for 5 min with 100 ng/ml VEGF and subsequently fixed in 4% PFA/4% sucrose supplemented with 1 mM sodium orthovanadate for 20 min at room temperature. Immunostaining with rabbit anti-(phospho) SFK (Y416) antibody followed by Alexa-488 conjugated secondary antibody was used to detect phosphorylated SFKs in the axonal growth cones and axons of the neurons. Phalloidin was used to visualize the actin

cytoskeleton. Image stacks were acquired at a confocal microscope (Zeiss LSM510, 63x objective) and fluorescence intensities of maximal intensity projections were quantified using FIJI. At least 38 growth cones and axons in three independent experiments were analyzed and statistical differences were assessed by unpaired students t test.

## Analysis of hippocampal gross morphology in vivo

20 µm thick cryosections from P10 pups were immune-stained for nuclei using TO-PRO3 (1:1000, Life technologies), neuronal nuclei using NeuN and for the axonal marker L1. Image stacks were obtained with a confocal microscope (Zeiss LSM510, 10x objective). To determine neuronal density, high magnification images of the NeuN stained CA3 region were acquired and the number of neuronal nuclei counted from three different brain sections and four animals per genotype. Based on the L1 staining, the dimensions of the hippocampal layers were determined by measuring the thickness of each layer and normalizing it to the total thickness of all layers as previously described (*Terauchi et al., 2010*).

## Analysis of axon branching in vivo

Control, *Nes-cre;Kdr^{lox/-}* and *Nes-cre;Kdr^{lox/+};EfnbB2^{lox/+}* pups sacrificed, the brains removed and embedded in 5% low-melting temperature agarose. 250 µm transverse sections were obtained using a vibratome (Leica) and collected in ice-cold PBS. The PBS was removed and sections fixed for 2 hr in 4% PFA at RT. Following three washes with PBS, single sections were transferred to a 12 well plate. four sections per brain and experiment were used. Under a stereomicroscope, all liquid was carefully removed with a P1000 pipette and a tiny DiI crystal (Molecular Probes, D-3911) was inserted into the CA3 pyramidal cell layer with the help of a tungsten needle attached to a glass pasteur pipette. The slices were covered with PBS and incubated at 37°C for 16–20 hr. Subsequently, the sections were stained with TO-PRO3 (1:1000, Life technologies) and mounted. Image stacks of Schaffer collaterals in the stratum radiatum of the CA1 region were acquired using a confocal microscope (Zeiss LSM510, 20x objective). Single axons were traced using the FIJI plugin Simple Neurite Tracer and the number and length of axon branches were determined.

## Transmission electron microscopy

For electron microscopy experiments, mice were perfused with 2.5% glutaraldehyde and 2% PFA. First, 400 µm thick vibratome section were cut, and those containing the hippocampus were collected. The vibratome sections were rinsed in cacodylate buffer (3–10 min) and then incubated in 0.5% $OsO_4$ in cacodylate buffer for 30 min. Afterwards the tissue was dehydrated in a series of increasing concentrations of acetone (30%, 50%, 70%, 85% and 2x 100% for 10 min each). The tissue was pre-incubated in a 1:1 mixture of acetone and Epon for 30 min, followed by incubation in pure Epon overnight at RT. On the next day the tissue was transferred into fresh Epon for polymerization for at least 48 hr in a heating cabinet at 60C. Vibratome sections were flat-embedded in Epon between two plastic sheets, cut out after polymerization and glued onto Epon blocks. Ultrathin sections were collected on slot grids, which were coated with 2% polyvinyl butyral in chloroform. An ultramicrotome (Ultracut UCT, Leica) with a diamond knife (Ultra 45, Diatome, Hatfield) was used to cut ultrathin sections at a thickness of 50-60 nm. The ultrathin sections were transferred with an eyelash onto the grids. Afterwards, the grids were counterstained with uranyl acetate and lead citrate using the automatic staining machine EM Stain (Leica). The transmission electron microscope (TEM) Leo 912 AB Omega (Zeiss) was used to acquire images from ultrathin sections. Images were taken by using a CCD camera (Troendle) in combination with the Image SP software (Troendle). Electron microscopy images of 20–50 µm from the middle region in stratum radiatum of the hippocampal CA1 region were acquired at a magnification of 5000x. At least eight images from different ultrathin sections were acquired per mouse, which corresponds to over 700 counted synapses per mouse. Synapses were identified and counted based on the presence of presynaptic vesicles, the synaptic cleft and postsynaptic density.

## Electrophysiology

### Hippocampal slice preparation

Transverse 400-um-thick hippocampal slices were made from pups (P8-P10) using a vibratome (Leica VT1200). For dissection and slicing, sucrose-based ACSF containing (in mM) 87 NaCl, 25 NaHCO$_3$, 2.5 KCl, 1.25 NaH$_2$PO$_4$, 10 glucose, 0.5 CaCl$_2$, 7 MgCl2, and 75 sucrose was used. For storage and superfusion of slices, ACSF solution containing (in mM) 125 NaCl, 25 NaHCO$_3$, 2.5 KCl, 1.25 NaH$_2$PO$_4$, 25 glucose, 2 CaCl$_2$, 1 MgCl$_2$, 2 TTX and 20 bicuculline was used. Slices were incubated at 34°C for 30 min after slicing and subsequently stored at room temperature. Both solutions were equilibrated with 95% O$_2$ and 5% CO$_2$ throughout the procedure.

### Electrophysiological recordings

Whole-cell patch-clamp recordings were made from CA1 pyramidal neurons with pipettes with a resistance of 4–6 MΩ using an Axopatch 200B amplifier. The internal solution of pipettes contained 120 mM potassium gluconate, 20 mM KCl, 0.1 mM EGTA, 2 mM MgCl$_2$, 4 mM Na$_2$ATP, 0.5 mM GTP, 10 mM HEPES, 7 mM disodium phosphocreatine, (pH 7.2, 300 mOsm). Neurons were voltage clamped at –70 mV while the series resistance was left uncompensated during the recordings. mEPSCs were analyzed offline using Stimfit software (*Guzman et al., 2014*) employing a template matching algorithm. Selected mEPSC events were individually screened with an amplitude threshold of >5 pA and an exponential decay. Recordings were started 5 min after patching and the minis are usually recorded for 10 min. Statistical differences between experimental conditions were determined by the unpaired students t-test. The experimenter was blind to the identity of the mice during acquisition.

## Preparation of neuronal cultures

Hippocampal neuron cultures were prepared from E16.5 mouse embryos as described previously (*Segura et al., 2007*) and cultured on coverslips or tissue culture plates coated with poly-L-lysine (1 mg/ml, Sigma) and laminin (20 µg/ml, Sigma). Neurons were maintained in neurobasal medium (Life technologies) supplemented with B27 (Invitrogen) and 0.5 mM Glutamine (GIBCO). For morphological analysis at 3 DIV neurons were cultured at a density of 20.000 neurons/coverslip.

## Neuronal transfection

For subcellular localization of the VEGFR2-GFP and tracking of actin dynamics, neurons were plated on four well Labtek chambers (Thermo Scientific, 155383) at a density of 100.000 neurons/well. At 1 DIV neurons were co-transfected with an hVEGFR2-GFP fusion plasmid and UtrCH-mCherry plasmid using Lipofectamine 2000 (Invitrogen) following manufacture instructions. mCherry-UtrCH plasmid was described in *Burkel et al. (2007)* and provided by Dr. Ulrike Engel (University of Heidelberg Heidelberg, Germany).

## Morphological analysis of neuronal cultures

1 DIV hippocampal neurons were stimulated for 48 hr with or without 100 ng/ml VEGF, fixed for 20 min in 4% PFA/4% sucrose at RT and stained for beta-III-tubulin. Where indicated, neurons were pre-treated with 2 µg/ml α-VEGFR2 blocking antibody (αVEGFR2 (DC101), gift from Dr. T Schmidt), anti-Nrp1 blocking antibody (αNrp1), respective IgG control, 1 µM PP2 (Calbiochem) or 1 µM PP3 (Calbiochem) for 2 hr prior to VEGF stimulation. The neurons were imaged using a Zeiss Axiovert 200 epifuorescence microscope. The longest neuronal process was defined as the axon and the number and length of axonal branches were quantified using the ImageJ plugin NeuronJ.

## Time-lapse video microscopy

For time-lapse video microscopy 50.000 neurons/well were plated in a 12 well plate. At 1 DIV neurons were pre-treated for 2 hr with 2 µg/ml α-VEGFR2 blocking antibody, respective IgG control, 1 µM PP2, 1 µM PP3. To block internalization, neurons were pre-treated with 80 µM dynasore or the equal volume of DMSO vehicle control for 30 min. Subsequently, 50 ng/ml VEGF was added and at least 10 neurons per condition were imaged simultaneously at the Nikon Imaging Center, Heidelberg, over the course of 4 hr at 4 min intervals using an inverted Nikon Ti microscope with Nikon perfect focus system and encased by an environmental box to keep temperature, CO$_2$ and humidity

constant. Phase contrast images were acquired using a Nikon Plan Fluor 10x NA 0.3 phase contrast objective. Newly formed and growing axon branches were defined as extension, while loss and shrinkage of branches were defined as retraction. The number and length of extending and retracting axon branches were quantified using the ImageJ plugin NeuronJ and the growth rates calculated. The number of newly formed filopodia was quantified within the first 80 min of the time-lapse movies.

## Dynamic VEGFR2-GFP movement and actin tracking

1 DIV hippocampal neurons were double transfected with VEGFR2-GFP and mCherry-UtrCH. At 3 DIV neurons were imaged using an inverted Nikon Ti microscope with objective TIRF illumination (Nikon Apo TIRF 60x NA 1.49) and equipped with an on-stage incubation chamber from TokiaHit for temperature, $CO_2$- and humidity control. GFP and mCherry were imaged sequentially with 488 nm and 561 nm excitation and emission filters 520/40 nm and 630/60 nm onto an EM-CCD (Andor Xion). To follow actin nucleation and patch formation, mCherry-UtrCH was imaged for 2 min at a frame rate of 1.6 fps before and 5 min after stimulation with 100 ng/ml VEGF. To investigate protrusion dynamics, mCherry-UtrCH images were acquired every 5 s over the course of 10 min before and after stimulation with 100 ng/ml VEGF. VEGFR2-GFP trafficking was observed for 1 min at a frame rate of 1.6 fps before and 5 min after stimulation with 100 ng/ml VEGF.

## Western Blotting

Neurons were harvested in ice-cold lysis buffer (150 mM NaCl, 20 mM Tris, 5 mM EDTA, 10% glycerol, 1% Igepal, protease and phosphatase inhibitor tablets (Roche)), lysed for 30 min on ice and cleared by centrifugation at 15.000 rpm for 15 min at 4C. Protein concentrations were determined using Roti-Quant Universal kit (Roth). Protein samples were loaded on 7.5% or 10% polyacrylamide gels for SDS-PAGE. Following transfer to 0.2 μm pore-size PVDF membranes, blots were blocked in 5% BSA in TBST for 1 hr at RT. Blots were then probed with antibodies against VEGFR2, (phospho-) VEGFR2 (Y1175), GFAP and VE-Cadherin. After incubation with appropriate HRP-conjugated secondary antibodies, bands were detected with ECL Western Blotting detection reagent (Millipore) and visualized using ImageQuant LAS 4000 (GE Healthcare).

## Reverse transcription PCR

Total RNA was prepared from neuronal and primary endothelial culture, as well as from postnatal brains using Superscript II (Invitrogen). Reverse transcription-PCR was carried out using Q5 High Fidelity DNA polymerase (New England Biolabs) according to manufacturer instruction. Forward (F) and reverse (R) primers used were as follows: *CD31:* F, 5'-AGCCTAGTGTGGAAGCCAAC-3'; R, 5'-AGCCTTCCGTTCTCTTGGTG-3'; *GAPDH:* F, 5'- GGTCCTCAGTGTAGCCCAAG-3'; R, 5'-AATGTG TCCGTCGTGGATCT-3'; *GFAP:* F, 5'-AAATCCGTGTCAGAAGGCCA-3'; R, 5'-TAATGACCTCACCA TCCCGC-3'; *Neuropilin 1:* F, 5'-AACGTGTGCTTCTGTCCAAC-3'; R, 5'-AAGGGAGAGGGGAAAG-CAAT-3'; *VEGF:* F, 5'-CGTTCACTGTGAGCCTTGTT-3'; R, 5'-CTTGGCTTGTCACATCTGCA-3'; *Vegfr1:* F, 5'-ACCCAGGAGTGCAAATGGAT-3'; R, 5'-TGTTGGACGTTGGCTTGAAG-3'; *Vegfr2 (Kdr):* F, 5'-TTCACAGTCGGGTTACAGGC-3'; R, 5'-CTGCCGACGTTCCTCTCTTT-3'.

## Quantitative RT-PCR

Total RNA was prepared from neuronal cultures and reverse transcription was performed using Superscript II (Invitrogen). Expression levels of VEGFR2 mRNA were quantified by real-time RT-PCR using SYBR-Mix (Applied Biosystems). GAPDH was served as an endogenous control. Following forward (F) and reverse primers (R) were used: *Vegfr2 (Kdr):* F, 5'- TTCACAGTCGGGTTACAGGC-3'; R, 5'- CTGCCGACGTTCCTCTCTTT-3'; *GAPDH:* F, 5'- GGTCCTCAGTGTAGCCCAAG-3'; R, 5'- AATG TGTCCGTCGTGGATCT-3'.

## Quantification of VEGF protein levels by ELISA

Conditioned medium of hippocampal neuronal cultures was collected at various times in vitro and processed for further measurements of VEGF levels using the Quantikine mouse VEGF ELISA kit (R and D, MMV00).

## Statistical analysis

Results are expressed as the mean ± SEM, unless stated otherwise. To calculate statistical significance the unpaired Student's t-test, one-way ANOVA followed by Tukey's multiple comparisons test and two-way ANOVA followed by Dunnett multiple comparisons test were calculated using Prism software. An excel file including all detailed statistical information of the data presented in the study is provided as a source data file.

## Acknowledgements

We thank all members of CRA and AAP labs for their help, critical inputs and fruitful discussions. We also acknowledge the Nikon Imaging Center of Heidelberg University for all their help and support, and Gong Sun Nam for excellent technical assistance. Research in CRA is supported by the Marie Curie Career integration grant (FP7-PEOPLE-2011-CIG-304050), by the European Research Council (ERC-StG-311367 NeuroVascular Link), by Deutsche Forschungsgemeinschaft (FOR2325; SFB1366 (Project number 394046768-SFB 1366); SFB873; SFB1158 and GRK2099). and by the Schram Foundation. Research in AAP is supported by the European Research Council (ERC_AdG_Neurovessel_project 669742), by the Deutsche Forschungsgemeinschaft (SFB 834, SFB1080, SFB1193, FOR2325, EXC 2026) and the Max Planck Fellow Program.

## Additional information

### Funding

| Funder | Grant reference number | Author |
|---|---|---|
| Deutsche Forschungsge-meinschaft | FOR2325 | Robert Luck<br>Carmen Ruiz de Almodóvar<br>Amparo Acker-Palmer |
| Schram Foundation | Individual grant | Carmen Ruiz de Almodóvar<br>Robert Luck<br>Andromachi Karakatsani |
| European Research Council | ERC-StG-311367 | Carmen Ruiz de Almodóvar |
| FP7 People: Marie-Curie Actions | FP7-PEOPLE-2011-CIG-304054 | Carmen Ruiz de Almodóvar |
| Deutsche Forschungsge-meinschaft | SFB1366 | Carmen Ruiz de Almodóvar |
| Deutsche Forschungsge-meinschaft | SFB873 | Carmen Ruiz de Almodóvar<br>Andromachi Karakatsani |
| Deutsche Forschungsge-meinschaft | SFB1158 | Carmen Ruiz de Almodóvar |
| Deutsche Forschungsge-meinschaft | GRK2099 | Carmen Ruiz de Almodóvar |
| European Research Council | ERC_AdG_Neurovessel_project 669742 | Amparo Acker-Palmer |
| Deutsche Forschungsge-meinschaft | SFB834 | Amparo Acker-Palmer |
| Deutsche Forschungsge-meinschaft | SFB1080 | Amparo Acker-Palmer |
| Deutsche Forschungsge-meinschaft | SFB1193 | Amparo Acker-Palmer |
| Deutsche Forschungsge-meinschaft | EXC 2026 | Amparo Acker-Palmer |

The funders had no role in study design, data collection and interpretation, or the decision to submit the work for publication.

## Author contributions
Robert Luck, Conceptualization, Formal analysis, Investigation, Methodology, Writing—original draft, Writing—review and editing; Severino Urban, Conceptualization, Investigation, Methodology, Writing—original draft; Andromachi Karakatsani, Investigation, Writing—review and editing; Eva Harde, Formal analysis, Investigation, Methodology; Sivakumar Sambandan, Investigation, Visualization; LaShae Nicholson, Silke Haverkamp, Rebecca Mann, Investigation; Ana Martin-Villalba, Resources, Methodology; Erin Margaret Schuman, Resources, Supervision; Amparo Acker-Palmer, Resources, Funding acquisition, Writing—review and editing; Carmen Ruiz de Almodóvar, Conceptualization, Resources, Supervision, Funding acquisition, Investigation, Writing—original draft, Project administration, Writing—review and editing

## Author ORCIDs
Ana Martin-Villalba (iD) http://orcid.org/0000-0002-9405-8910
Erin Margaret Schuman (iD) https://orcid.org/0000-0002-7053-1005
Carmen Ruiz de Almodóvar (iD) https://orcid.org/0000-0001-5975-7815

## Ethics
Animal experimentation: All animal experiments were conducted in accordance with the German institutional guidelines and ethical committees (references refs: FU1090, T60/12, T79/13, T49/15, T69/14, T46/16, T36/17, T48/18, and I19/13).

## Decision letter and Author response
Decision letter https://doi.org/10.7554/eLife.49818.sa1
Author response https://doi.org/10.7554/eLife.49818.sa2

# Additional files

## Supplementary files
• Transparent reporting form

## Data availability
All data generated or analysed during this study are included in the manuscript and supporting files. Source data files have been provided.

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
