## [Decision Letter]

**Acceptance summary:**

It is known that VEGF-A signaling plays a certain role neuronal development. In this study, the authors uncover molecular and mechanical insights into how neuronal VEGF-A-VEGFR2 regulates axonal branching and synapse formation in the different subsets of hippocampal neurons. Overall the study is intriguing and the results are rigorous and solid.

**Decision letter after peer review:**

Thank you for submitting your article "VEGFR2 controls axon branching of hippocampal neurons during development" for consideration by *eLife*. Your article has been reviewed by Didier Stainier as the Senior Editor, a Reviewing Editor, and three reviewers. The following individuals involved in review of your submission have agreed to reveal their identity: Injune Kim (Reviewer #1); Bassem A Hassan (Reviewer #2).

The reviewers have discussed the reviews with one another and the Reviewing Editor has drafted this decision to help you prepare a revised submission.

The comments are relatively favorable, complimentary and constructive. However, the authors are required to address following.

1) It is essential to further clarify the complexities of VEGFR2/VEGFR2 ligands – VEGF-A, C and D in the axon branching using a primary neuron culture.

2) Please address the points of all three reviewers point-by-point in a data-driven manner or with further analyses.

I believe the authors are capable of addressing most of the comments, but please provide the reasons for not implementing the suggested changes where necessary.

*Reviewer #1:*

This work elucidated the role of the VEGF signaling in CA3 hippocampal axon branching. Although the experiments were designed and performed well, the complex behavior of axon responding to VEGF activation and the loss of VEGFR2 remains to be clarified. This issue needs to be further discussed for improving the manuscript.

1) Together with the accompanying manuscript by Harde et al., this work shows the opposite effect of the loss of VEGFR2 on the branching of axons and dendrites in hippocampal neurons. How does the axon sense the VEGF cue in the absence of VEGFR2? What about other VEGF receptors?

2) Does the VEGFR2 interact with EphrinB2 for its internalization in the axon?

3) Hippocampal neurons appear to be in proximity of blood vessels. Does the loss of neuronal VEGFR2 gene affect angiogenesis in neighboring vessels due to the potential increase of free VEGF level?

*Reviewer #2:*

Urban et al. present convincing and clear evidence that neuronal VEGF signaling via the VEGFR2 receptor regulates axonal branching. In an interesting twist both gain and loss of function of VEGFR2 increase axonal branching, but with what appear to be opposite consequences on synapse formation.

Overall, the manuscript is very well written, the evidence clear and convincing, and interpretation consistent with the data. In principle, the manuscript can be published in *eLife* in its current form. I do, however have some suggestion for the authors, which I think would add the "completeness" of the story.

1) The authors show that loss of VEGFR2 reduces the number of synapses. Perhaps I missed this, but I couldn't find data as what happens to synapses upon stimulation with VEGF. Do they increase in number, as would be predicted from the increase in what appear to be mature branches?

2) In reference to the accompanying manuscript, is VEGFR2 internalization also required for its role in axonal branching?

3) Very recent work from the Hiesinger lab (Ozel et al., Dev Cell 2019) shows how filopodial dynamics regulate synapse formation in Drosophila. Given the role of VEGFR2 shown here in actin regulation and branching, discussing potential regulation of filopodial dynamics by VEGFR2 in this context might be interesting.

4) In my personal opinion terms like "X controls Y process" are outdated and reflect linear deterministic models that are no longer relevant in modern biology and should in my view no longer be used. Perhaps a term like "regulates" or "contributes to" are more appropriate in the title, but I leave that to the discretion of the authors.

*Reviewer #3:*

The manuscript by Urban et al. investigates the role of VEGF-A signaling in CA3 neurons in the hippocampus with regard to axon branching. Expression of both ligand and receptor VEGFR2 is investigated, then primary culture of neurons is used to show that VEGF stimulation leads to excess branching. However, in vivo the neuron-specific conditional genetic loss of the receptor also leads to excess branches, although these are thought to be less mature. A blocking antibody to VEGFR2 recapitulates the excess branching seen in vivo in primary cultures. The signaling is linked to Src-like kinases using inhibitors, and some very elegant cell biology indicates that VEGFR2 relocalizes to actin patches upon stimulation.

Overall the data is rigorous and uses both in vitro and in vivo approaches. It is known that VEGF-A signaling is important in neuronal development. The major novelty here is that the neuronal loss of VEGFR2 leads to excess branching of a subset of hippocampal neurons that does not appear to lead to productive synapse formation, while VEGF-A stimulation in primary neurons also leads to excess branching. The authors speculate that the lack of synapses may lead to a feedback loop with more short axon branches. It would strengthen the work to test some other ideas – for example perhaps VEGF-A is not the relevant ligand in vivo since VEGF-C and VEGF-D also bind to VEGFR2. What are the effects of alternate ligands on the parameters in primary neuron culture?

The other aspect of the novelty is the idea that VEGFR2 relocalizes upon VEGF-A stimulation to actin patches, where it supposedly promotes neurite outgrowth. It would strengthen the work to move the kymographs to the main figures and label better to highlight these novel findings, and if possible, to include the movies from which the kymographs were derived (the co-labeling of actin and VEGFR2). Also, would strengthen to show spatial changes by a different approach – is it possible to use Proximity Ligation Assay or pull down to investigate VEGFR2 association with something localized to the actin patches and/or filopodia?

Also, as described in review of co-submitted paper, it would be useful to better integrate the findings from the co-submitted paper in the Discussion section – it appears that endocytosis is important for dendrite biology, is it also important for the axon branching effects of VEGF/VEGFR2? Are SPKs important for dendrites as well as axons? It is not clear whether axons and dendrites use different mechanisms or whether they were tested differently.

---

## [Author Response]

The comments are relatively favorable, complimentary and constructive. However, the authors are required to address following.1) It is essential to further clarify the complexities of VEGFR2/VEGFR2 ligands – VEGF-A, C and D in the axon branching using a primary neuron culture.

It is indeed plausible that VEGF-C and VEGF-D might contribute to and affect axonal branching in the hippocampus during development. To address this concern we checked their expression pattern in the developing hippocampus as well as their effect in primary hippocampal neuron cultures. For detailed description and interpretation of the new results please see our response to reviewer #3, comment #1.

2) Please address the points of all three reviewers point-by-point in a data-driven manner or with further analyses.

In our response letter we address point-by-point all reviewers’ comments.

I believe the authors are capable of addressing most of the comments, but please provide the reasons for not implementing the suggested changes where necessary.Reviewer #1:This work elucidated the role of the VEGF signaling in CA3 hippocampal axon branching. Although the experiments were designed and performed well, the complex behavior of axon responding to VEGF activation and the loss of VEGFR2 remains to be clarified. This issue needs to be further discussed for improving the manuscript.

We thank the reviewer for his comments. Below we have addressed point by point these general concerns. In addition, as indicated by this reviewer, in the Discussion section of our manuscript, we have further discussed the complex behavior of axons upon activation as wells as loss of VEGFR2.

1) Together with the accompanying manuscript by Harde et al., this work shows the opposite effect of the loss of VEGFR2 on the branching of axons and dendrites in hippocampal neurons. How does the axon sense the VEGF cue in the absence of VEGFR2? What about other VEGF receptors?

We thank the reviewer for his comment. In the submitted manuscript, we show that stimulation of hippocampal neurons with VEGF increased axon branching via activation of its cognate receptor VEGFR2. At the same time, blockage, as well as knockout of VEGFR2 showed the same phenotype of increased axonal branching, both in vitro and in vivo (in *Nes-cre;Kdr^lox/-^* mice). In our manuscript we speculate that the increased axon branching observed upon loss of VEGFR2 is a compensatory mechanism. In this regard, as this reviewer suggested, it might be possible that interference with VEGFR2 function shifts the signaling of VEGF to other VEGF receptors, thus in turn being the reason for the increase in axon branching. Equally likely, the compensatory mechanism that is activated could be totally independent of VEGF receptor signaling.

To address whether VEGF could be binding to other VEGF receptors in the absence of VEGFR2 we focused on VEGFR1 (Flt1) and Neuropilin 1 (Nrp1), the other two receptors via which VEGF can induce a signal transduction cascade independent of VEGFR2 (de Vries et al., 1991; Soker et al., 1998; Roth et al., 2016).

VEGFR1

We have analyzed the expression of *Vegfr1* in the hippocampus at similar time points as we did for VEGFR2 (late embryonic (E18.5) and early postnatal stages P4 and P8) using ISH. At these time points *Vegfr1* mRNA is primarily detected in blood vessel structures (Author response image 1). Similar to our results, the data provided by the Allen Brain Atlas for the developing mouse brain (http://developingmouse.brain-map.org) shows *Vegfr1* mRNA expression primarily in blood vessels in the hippocampus (Author response image 1).

**Author response image 1. respfig1:** Flt1 is expressed by blood vessels during hippocampal development. (**A, B**) *Flt1*-ISH was performed in the mouse hippocampus at E18.5, P4 and P8 developmental stages. Antisense (AS, in **A**) and sense (SE, in **B**) signals are shown. CA1 and CA3 hippocampal areas, as well as the DG are indicated. Lower panel in A) shows higher magnification inserts. Scale bars 500 µm (**A** – higher panel and **B**) and 250 µm (**A** – lower panel). C) *Flt1*-ISH from the Allen Brain Atlas, showing AS signal in the hippocampus at E18.5, P4 and P14. Scale bars 500 µm.

These data also correlates with our analysis shown in Figure 1—figure supplement 1D of our revised manuscript where no expression of *Vegfr1* mRNA is detected in isolated hippocampal neurons at 1 DIV and 3 DIV. Based on this expression pattern we believe that VEGFR1 is a rather improbable receptor for VEGF in hippocampal neurons during development.

Nrp1

Nrp1 is a known mediator of the repulsive effect exerted by Semaphorins on axon development, which is also specifically described for the hippocampus (He and Tessier-Lavigne, 1997; Chedotal et al., 1998). Nrp1 is also described to function as co-receptor for VEGFR2 as well as to transduce VEGF signals independent of VEGFR2 (Soker et al., 1998; Roth et al., 2016; Erskine et al., 2011). To investigate whether Nrp1 could act as a receptor for VEGF and regulate axonal branching in hippocampal neurons, we first analyzed its expression pattern during hippocampal development using ISH. We detected *Nrp1* mRNA in the CA1, CA2 and CA3 hippocampal regions, as well as in the dentate gyrus at late embryonic (E18.5) and early postnatal (P4 and P8) stages (Figure 5—figure supplement 4A of our revised manuscript). This is in line with our expression analysis in isolated hippocampal neurons (1 DIV and 3 DIV; Figure 1—figure supplement 1D of our revised manuscript) and with previous publications showing the expression of *Nrp1* mRNA in the hippocampus and neocortex during development (Chedotal et al., 1998; Kawakami et al., 1996). The spatiotemporal expression of *Nrp1* is similar to the one of *Vegfr2* and *Vegf*, raising the possibility that Nrp1 can act either alone or together with VEGFR2 to mediate VEGF signaling in the developing hippocampus.

First, to elucidate whether VEGF could also signal via Nrp1 to modulate axon branching, we blocked Nrp1 using a previously described α-Nrp1-neutralizing antibody (Ruiz de Almodovar et al., 2011) and analyzed the VEGF-mediated effect on axonal branching as before. While axon branch number and length was significantly increased in control neurons upon VEGF stimulation, this effect was not blocked in the presence of α-Nrp1-neutralizing antibody (Figure 5—figure supplement 4B and C of our revised manuscript). This result suggests that Nrp1 is dispensable for VEGF-induced axon branching.

Second, we also considered the possibility that the increase in axon branching upon α-VEGFR2 treatment could be due to a swift towards VEGF-Nrp1 signaling. To test this hypothesis, we co-administered α-Nrp1-neutralizing antibody together with αVEGFR2 in hippocampal neurons and analyzed axon branching in the different conditions. Consistent with our data, α-VEGFR2 led to an increase in axon branch number and length (Figure 5—figure supplement 4D and C of revised manuscript). This effect was however not blocked with the addition of the α-Nrp1-neutralizing antibody (Figure 5—figure supplement 4D and C of revised manuscript), indicating that Nrp1 is not the receptor that leads to increased axon branching upon inactivation of VEGFR2.

Collectively, these data show that Nrp1 does not mediate the effect of VEGF in axon branching by itself, nor does it represent a strong compensatory mechanism that takes over when VEGFR2 is blocked. Our data also suggests that this compensatory mechanism might be independent of VEGF signaling. In the revised version of our manuscript we have now integrated the data for Nrp1 in Figure 5—figure supplement 4 of the revised manuscript.

2) Does the VEGFR2 interact with EphrinB2 for its internalization in the axon?

To address the reviewer’s question of whether VEGFR2 interacts with EphrinB2 for its internalization in the hippocampal axon, similar as it occurs in dendrites (shown in the accompanying manuscript by Harde et al., we isolated hippocampal neurons from EphrinB2 knockout mice (from here on EphrinB2^-/-^) and analyzed the number and length of axonal branches in the presence and absence of VEGF. Stimulation of hippocampal neurons from EphrinB2^-/-^ with VEGF resulted in a similar increase of axon branch number and length as in neurons from control littermates (Figure 6—figure supplement 1C and D of revised manuscript), indicating that EphrinB2 is not necessary for the VEGF-induced effect on axonal branching.

To confirm this in vitro data, we further analyzed CA3 hippocampal axon branching in vivo in VEGFR2 and EphrinB2 compound mice (from here on *Nes-cre;Kdr^+/-;^ephrinB2^+/-^*), which showed a phenotype in dendritic branching in the accompanying paper (Harde et al.,) Using the same DiI-labeling and quantification approach we used for analyzing CA3 axon branching in *Nes-cre;Kdr^lox/-^*, we labeled single axons of CA3 pyramidal neurons in wildtype and *Nes-cre;Kdr^+/-^;ephrinB2^+/-^* littermates and analyzed their branching. This analysis revealed no significant differences between the genotypes (Figure 6—figure supplement 1E of revised manuscript), indicating that the interaction between VEGFR2 and EprhinB2 is not necessary for the VEGF-mediated regulation of hippocampal axon branching.

Altogether, these new results show that VEGF utilizes different receptor complexes and molecular mechanisms to regulate axon and dendritic branching in hippocampal neurons. The new data generated with EphrinB2 are now integrated in the updated version of our manuscript (Figure 6—figure supplement 1 of revised manuscript).

3) Hippocampal neurons appear to be in proximity of blood vessels. Does the loss of neuronal VEGFR2 gene affect angiogenesis in neighboring vessels due to the potential increase of free VEGF level?

We understand the reviewer’s reasoning that upon loss of neuronal VEGFR2 expression there might be potential increase of free VEGF level, which subsequently might affect angiogenesis in neighboring vessels. A similar mechanism has been described before in the developing mouse retina (Okabe et al., 2014). To address this question, we analyzed the area covered by blood vessels in the hippocampus of *Nes-cre;Kdr^lox/-^* mice at P10 using ImageJ. We did not observe a significant difference in blood vessel density in the hippocampus of *Nes-cre;Kdr^lox/-^* mice when compared to their control littermates (Author response image 2).

**Author response image 2. respfig2:** Neuronal loss of VEGFR2 does not affect blood vessel density in the hippocampus. (**A**) Blood vessels were stained using Isolectin-B4 at P10 in control and *Nes-cre;Kdr^lox/-^*animals. The area of Isolectin-B4-positive signal was quantified using ImageJ and normalized to the hippocampal area. Data are represented as mean ± SEM, >15 neurons from n=4 animals. n.s. not significant; unpaired Student’s t-test.

Reviewer #2:Urban et al. present convincing and clear evidence that neuronal VEGF signaling via the VEGFR2 receptor regulates axonal branching. In an interesting twist both gain and loss of function of VEGFR2 increase axonal branching, but with what appear to be opposite consequences on synapse formation.Overall, the manuscript is very well written, the evidence clear and convincing, and interpretation consistent with the data. In principle, the manuscript can be published in eLife in its current form.

We thank the reviewer for their positive comments.

1) The authors show that loss of VEGFR2 reduces the number of synapses. Perhaps I missed this, but I couldn't find data as what happens to synapses upon stimulation with VEGF. Do they increase in number, as would be predicted from the increase in what appear to be mature branches?

Indeed, in Figure 5 of our manuscript we show that despite the increase in axon branching upon loss of VEGFR2 in neurons, the number of functional synapses in the CA1 is reduced, suggesting that those branches are rather immature and/or nonfunctional. In our manuscript we did not analyze synapse formation in the axon upon stimulation with VEGF.

In the accompanying manuscript of Harde et al., the authors investigated the effect of VEGF on the dendritic spines of hippocampal neurons in vitro (which would be the postsynaptic side for the axon). They transfected primary hippocampal neurons from wildtype mice with a plasmid expressing EGFP at 11 DIV, stimulated the neurons with 50 ng/ml VEGF for 24h or 48h and analyzed spine morphology and number at 14 DIV. Quantification revealed a significant increase in the number of mature spines upon VEGF stimulation when compared to unstimulated control neurons. The authors further show that the effect of VEGF is mediated through activation of its receptor VEGFR2.

Previous studies have also attributed a role for VEGF in regulating synapse formation and synaptic plasticity in the adult hippocampus (De Rossi et al., 2016; Kim et al., 2008; Latzer et al., 2019; Licht et al., 2011). De Rossi et al. for example demonstrated that VEGF signaling via VEGFR2 increases the postsynaptic responses mediated by a specific type of glutamate receptors (GluNRs) in hippocampal neurons, thus highlighting the potential of VEGF signaling in neurons for regulating proper synaptic function (De Rossi et al., 2016). Using conditional transgenic systems to reversibly overexpress VEGF or block endogenous VEGF in the hippocampus of adult mice, Licht et al. demonstrated a crucial role for VEGF in modulating plasticity of mature neurons and subsequently regulating/affecting memory formation and LTP responses (Licht et al., 2011).

Altogether, although we did not specifically analyzed synapses in the axon upon VEGF stimulation, the above-mentioned studies point to a role for VEGF in synapse formation and plasticity.2) In reference to the accompanying manuscript, is VEGFR2 internalization also required for its role in axonal branching?

To investigate whether VEGFR2 internalization is required for its role in axonal branching, we followed a similar approach as in the accompanying manuscript (Harde et al.). We inhibited VEGFR2 internalization with dynasore, a dynamin inhibitor that 6 blocks both clathrin-dependent and clathrin-independent endocytic pathways, and analyzed the VEGF-mediated effect on axonal branching. Hippocampal neurons from wildtype mice were pretreated with dynasore or DMSO (as control) at 1 DIV for 30 min, followed by stimulation with 50 ng/ml VEGF. Time-lapse videos were recorded over the course of 4h and the dynamics of axon branching was analyzed. Treatment with dynasore blocked the increase in branch number and length induced by VEGF when compared to control conditions (DMSO treated neurons) (Figure 6—figure supplement 1A-B of revised manuscript).

These results indicate that internalization of VEGFR2 is also required for the VEGF-mediated effect on axon branching. However, in contrast to what occurs for dendritic branching, the internalization of VEGFR2 required for regulating axon branching is not dependent of EphrinB2 as we did not observed any significant inhibition of VEGF-induced axon branching in primary hippocampal neurons isolated from EphrinB2 knockout mice (see response to reviewer #1, comment 2 and Figure 6—figure supplement 1C-E of revised manuscript).3) Very recent work from the Hiesinger lab (Ozel et al., Dev Cell 2019) shows how filopodial dynamics regulate synapse formation in Drosophila. Given the role of VEGFR2 shown here in actin regulation and branching, discussing potential regulation of filopodial dynamics by VEGFR2 in this context might be interesting.We thank the reviewer for pointing out this interesting study to us. Indeed, we show that loss of VEGFR2 changes filopodia dynamics resulting in an increased number of filopodia and increased axon branching. With this in mind and considering that we also observe a reduced number of functional synapses on CA1 pyramidal neurons, it could very well be the case that the presence of VEGFR2 is somehow required for the competitive distribution of synaptic building material between synaptogenic filopodia. If this is the case, based on our data, we could speculate that in the absence of VEGFR2 the synaptic building material is reduced or distributed aberrantly, allowing only the formation and maturation of a reduced number of synapses. As in its current stage this is all speculation, and perhaps a bit far from the focus of our study, we prefer not to discuss it in our manuscript, but will consider following up on this possibility in further studies.4) In my personal opinion terms like "X controls Y process" are outdated and reflect linear deterministic models that are no longer relevant in modern biology and should in my view no longer be used. Perhaps a term like "regulates" or "contributes to" are more appropriate in the title, but I leave that to the discretion of the authors.

We thank the reviewer’s suggestion. We have modified the title accordingly and it now reads: “VEGF/VEGFR2 signaling regulates hippocampal axon branching during development.”

Reviewer #3:The manuscript by Urban et al. investigates the role of VEGF-A signaling in CA3 neurons in the hippocampus with regard to axon branching. Expression of both ligand and receptor VEGFR2 is investigated, then primary culture of neurons is used to show that VEGF stimulation leads to excess branching. However, in vivo the neuron-specific conditional genetic loss of the receptor also leads to excess branches, although these are thought to be less mature. A blocking antibody to VEGFR2 recapitulates the excess branching seen in vivo in primary cultures. The signaling is linked to Src-like kinases using inhibitors, and some very elegant cell biology indicates that VEGFR2 relocalizes to actin patches upon stimulation.Overall the data is rigorous and uses both in vitro and in vivo approaches.

We thank the reviewer for their positive comments. Below we have addressed point by point their general concerns.

It is known that VEGF-A signaling is important in neuronal development. The major novelty here is that the neuronal loss of VEGFR2 leads to excess branching of a subset of hippocampal neurons that does not appear to lead to productive synapse formation, while VEGF-A stimulation in primary neurons also leads to excess branching. The authors speculate that the lack of synapses may lead to a feedback loop with more short axon branches. It would strengthen the work to test some other ideas – for example perhaps VEGF-A is not the relevant ligand in vivo since VEGF-C and VEGF-D also bind to VEGFR2. What are the effects of alternate ligands on the parameters in primary neuron culture?

To address this question, we performed a thorough literature research on the expression and role of VEGF-C and VEGF-D ligands in the developing hippocampus. We also analyzed expression of VEGF-C and VEGF-D in the developing hippocampus ourselves and tested their effect on axon branching of isolated hippocampal neurons.

VEGF-C

VEGF-C binds primarily to VEGFR3 but it can also bind VEGFR2 after proper proteolytic cleavage (Joukov et al., 1996), leading to the formation and activation of VEGFR2/VEGFR3 heterodimers (Tammela and Alitalo, 2010; Koch and Claesson-Welsh, 2012; Thomas et al., 2013). Ward et al. described VEGF-C expression during embryonic and postnatal development in various brain regions (Ward and Cunningham, 2015). In the hippocampus, VEGF-C expression appears to be higher in embryonic stage E16 and declines in the maturing hippocampus, where it becomes restricted to the hilus and the granule cell layer (GCL) of the dentate gyrus (DG) (Ward and Cunningham, 2015).

To confirm that VEGF-C is expressed in the developing hippocampus, as indicated by Ward and Cunningham (2015), we micro-dissected the hippocampus from the developing mouse brain at P4 and P8 and performed an ELISA for VEGF-C. As positive controls we used the adult mouse heart and lung as VEGF-C is expressed in those tissues (Kukk et al., 1996). In contrast to Ward et al., we did not detect VEGF-C in the developing hippocampus at any of the analyzed stages, while its expression was positive in heart and lung (Author response image 3). The difference between our results and Ward et al. could be potentially explained by the sensitivity of the different approaches used to detect VEGF-C protein.

Nevertheless, to not exclude the possibility that VEGF-C is expressed in low (undetected by ELISA) levels and thus could exert an effect in developing hippocampal neurons, we cultured primary hippocampal neurons and stimulated them with 100 ng/ml VEGF-C at 1 DIV over the course of 48 hours. Analysis of axonal morphology in response to VEGF-C stimulation from three independent experiments showed no significant increase of axon branch number or length when compared to control or VEGF (VEGF-A) stimulated neurons (Author response image 3).

**Author response image 3. respfig3:** VEGF-C and VEGF-D do not show the same growth promoting effect on hippocampal axon branches as VEGF-A. (**A, B**) Hippocampal tissue was collected from mice at P4 and P8 stages. Heart and lung tissue was collected from adult mice. The concentration of VEGF-C (**A**) and VEGF-D (**B**) was measured in whole tissue lysates using ELISA. Data are represented as mean ± SEM, n>2 independent animals. nd not detectable. (**C, D**) 1 DIV hippocampal neurons were stimulated with 100 ng/ml VEGF-C, VEGF-D, VEGF-A or vehicle control for 48h. The number of axonal branches (**C**) and the average branch length (**D**) were analyzed. Data are represented as% of non-stimulated control. Mean ± SEM, >79 neurons from n=3 independent experiments. *p<0.05; **p<0.001; ***p<0.0001; one-way ANOVA.

VEGF-D

VEGF-D can also bind and activate VEGFR2 and VEGFR3 receptor (Achen et al., 1998; Davydova et al., 2016). VEGF-D expression has been described in the hippocampus in early postnatal and adult stages where its expression is regulated by neuronal activity and nuclear calcium signaling (Mauceri et al., 2011). Mauceri et al. showed that VEGF-D promotes dendritic maintenance, both in cultured hippocampal neurons as well as in the adult mouse hippocampus (Mauceri et al., 2011). Indeed, our own analysis of VEGF-D expression via ELISA also confirmed those previous results. As VEGF-D and its receptors VEGFR2 and VEGFR3 are expressed in the hippocampus, we set out to investigate a possible effect of VEGF-D on axonal development. We cultured primary hippocampal neurons and stimulated them with 100 ng/ml VEGF-D at 1 DIV over the course of 48hours (neurons from three independent experiments) and analyzed the axonal morphology. VEGF-D stimulation of primary hippocampal neurons did not lead to an increase in the branch number when compared to control and VEGF (VEGF-A) stimulated neurons. VEGF-D however induced a significant increase in the length of axon branches.

Altogether, our new results point to a potential effect of VEGF-D in regulating the length of hippocampal axon branches. Still, VEGF-A seems to be the main factor that, acting via VEGFR2, leads to an increase number of axon branches. We believe that describing the entire mechanism of how VEGF-D regulates axon branch growth, and whether it does so via VEGFR2 activation, would require substantially more detailed experiments, which could be performed in a follow up study. For this reason, we prefer to answer this part of the reviewer’s comment just in the response letter and not include the data in the current manuscript.

The other aspect of the novelty is the idea that VEGFR2 relocalizes upon VEGF-A stimulation to actin patches, where it supposedly promotes neurite outgrowth. It would strengthen the work to move the kymographs to the main figures and label better to highlight these novel findings, and if possible, to include the movies from which the kymographs were derived (the co-labeling of actin and VEGFR2).

We thank the reviewer for their suggestion. We have now moved the kymographs to the main figure (Figure 4 of revised manuscript) and included also the movies from which the kymographs were derived (Video 5, Video 6, Video 7 and Video 8).

Also, would strengthen to show spatial changes by a different approach – is it possible to use Proximity Ligation Assay or pull down to investigate VEGFR2 association with something localized to the actin patches and/or filopodia?

In the submitted manuscript we indeed investigated the specific localization and dynamics of VEGFR2 along the axon, as well as its potential association with actin patches and filopodia (Figure 4—figure supplement 1B-D of revised manuscript). For this aim, the approach we chose to follow was co-transfection of primary hippocampal neurons with a VEGFR2-GFP and a mCherry-UtrCH plasmid. This decision was based on the fact that (i) Utrophin is frequently used as a marker for stable and dynamic F-actin, both in fixed and living cells (Burkel, von Dassow and Bement, 2007; Melak, Plessner and Grosse, 2007); and that (ii) the calponin homology domains of utrophin interacts with adjacent actin subunits, thus further confirming that utrophin is a reliable marker for actin filaments/patches (Lin at al., 2007).

Our analysis confirmed the co-localization of VEGFR2 with utrophin in primary hippocampal neurons in vitro, suggesting an association between VEGFR2 and the actin cytoskeleton and a role of VEGFR2 in actin remodeling (Figure 4A). We therefore believe that the approach chosen is appropriate to show VEGFR2 localization in actin patches/filopodia.

In support of such an association, the direct effect of VEGFR2 on the actin cytoskeleton has been previously described in the context of angiogenesis, where VEGFR2 signaling leads to actin remodeling by phosphorylation of profilin-1 (Pfn-1), a small actin-binding protein, either directly or through Src tyrosine kinases (Fan et al., 2012). Actin remodeling mediated via VEGF/VEGFR2 signaling has also been described in the neuronal growth cones of chicken dorsal root ganglia (Olbrich et al., 2007).

Taken together, these data support a role of VEGFR2 in regulating actin dynamics and strengthen our results on a direct implication of VEGF/VEGFR2 signaling on actin remodeling in the axons hippocampal neurons.

Also, as described in review of co-submitted paper, it would be useful to better integrate the findings from the co-submitted paper in the Discussion section – it appears that endocytosis is important for dendrite biology, is it also important for the axon branching effects of VEGF/VEGFR2?

In the revised version of the manuscript, we have better integrated the findings from the co-submitted paper in the Discussion section.

We also checked whether internationalization of VEGFR2 is required for axon branching. For this, we followed a similar approach as in the accompanying manuscript (Harde et al.). We inhibited VEGFR2 internalization with dynasore, a dynamin inhibitor that blocks both clathrin-dependent and clathrin-independent endocytic pathways, and analyzed the VEGF-mediated effect on axonal branching. Hippocampal neurons from wildtype mice were pretreated with dynasore or DMSO (as control) at 1 DIV for 30 min, followed by stimulation with 50 ng/ml VEGF. Time-lapse videos were recorded over the course of 4h and the dynamics of axon branching was analyzed. Treatment with dynasore blocked the increase in branch number and length induced by VEGF when compared to control conditions (DMSO treated neurons) (Figure 6—figure supplement 1A and B of revised manuscript).

These results indicate that internalization of VEGFR2 is required for the VEGF-mediated effect in axon branching. However, in contrast to what occurs for dendritic bragf-aching, the internalization of VEGFR2 required for regulating axon branching is not dependent of EphrinB2 as we did not observed any significant inhibition of VEGF-induced axon branching in primary hippocampal neurons isolated from EphrinB2 knockout mice (see response reviewer 1; comment 2; and Figure 6—figure supplement 1C-E of revised manuscript).

Are SPKs important for dendrites as well as axons? It is not clear whether axons and dendrites use different mechanisms or whether they were tested differently.

We assume that the reviewer meant to ask whether SFKs are important for dendrites as well as for axons. In the accompanying paper, Harde et al. show that VEGF stimulation of 14 DIV hippocampal neurons induces SFKs activation (Figure 3C, accompanying manuscript). As SFKs activity is required for EphrinB2 mediated spine morphogenesis (Segura et al., 2007), and as EphrinB2 is also required for VEGF/VEGFR2-mediated regulation of dendritogenesis and spine morphogenesis in hippocampal neurons, we assume that indeed SFK activity is also required in the context of VEGF/VEGFR2induced dendritogenesis and spinogenesis.